# Hecw controls oogenesis and neuronal homeostasis by promoting the liquid state of ribonucleoprotein particles

Valentina Fajner [1], Fabio Giavazzi [2], Simona Sala [1], Amanda Oldani[1], Emanuele Martini [1], Francesco Napoletano [3,4], Dario Parazzoli [1], Giuliana Cesare[5], Roberto Cerbino [2,6], Elena Maspero [1✉], Thomas Vaccari [5✉] & Simona Polo [1,7✉]

Specialised ribonucleoprotein (RNP) granules are a hallmark of polarized cells, like neurons and germ cells. Among their main functions is the spatial and temporal modulation of the activity of specific mRNA transcripts that allow specification of primary embryonic axes. While RNPs composition and role are well established, their regulation is poorly defined. Here, we demonstrate that Hecw, a newly identified *Drosophila* ubiquitin ligase, is a key modulator of RNPs in oogenesis and neurons. Hecw depletion leads to the formation of enlarged granules that transition from a liquid to a gel-like state. Loss of Hecw activity results in defective oogenesis, premature aging and climbing defects associated with neuronal loss. At the molecular level, reduced ubiquitination of the Fmrp impairs its translational repressor activity, resulting in altered Orb expression in nurse cells and Profilin in neurons.

[1] IFOM, Fondazione Istituto FIRC di Oncologia Molecolare, Milan, Italy. [2] Dipartimento di Biotecnologie Mediche e Medicina Traslazionale, Università degli Studi di Milano, Segrate, Italy. [3] Laboratorio Nazionale CIB, Area Science Park, Padriciano 99, Trieste, Italy. [4] Dipartimento di Scienze della Vita, Università degli Studi di Trieste, Trieste, Italy. [5] Dipartimento di Bioscienze, Università degli Studi di Milano, Via Celoria 26, Milan, Italy. [6] Present address: Faculty of Physics, University of Vienna, Boltzmanngasse 5, Vienna, Austria. [7] Dipartimento di Oncologia ed Emato-oncologia, Università degli Studi di Milano, Milan, Italy. ✉email: elena.maspero@ifom.eu; thomas.vaccari@unimi.it; simona.polo@ifom.eu

RNP granules play a fundamental role in both germ and somatic cells as they are responsible for the mRNA transport and local translation required for neuronal and oocyte maturation[1–3]. Oogenesis has been intensively studied in *Drosophila* where many genes essential for germline development belong to these membrane-less granules[4]. Mutations in critical RNP components[5,6] have been linked to neurological diseases such as amyotrophic lateral sclerosis (ALS) and frontotemporal dementia (FTD), highlighting the crucial contributions of RNA localisation and translational control to long-term neuronal integrity[7,8]. Furthermore, liquid-to-solid–state transitions are emerging as the key events in the formation of intracellular pathological protein aggregates[5,9,10]. Whereas the ubiquitin-proteasome system has been demonstrated to play a major role in pathological aggregates, our understanding of the role/s played by the ubiquitin system in the RNPs metabolism and dynamics is limited[11].

In the ubiquitination cascade, E3 ligases act as catalysts and molecular matchmakers of the reaction. They carry out the final step of covalently binding ubiquitin (Ub) to substrates, usually to lysine (K) residues, and define the type of Ub chains attached to them, thus dictating the targets' fate[12]. Among the HECT (Homologous to the E6-AP Carboxyl Terminus) E3s families that are present in all eukaryotes, NEDD4 is the most characterised, with one member in yeast, three in *Drosophila melanogaster* and nine in humans. NEDD4 ligases modify multiple substrates by monoubiquitination[13,14] or by addition of K63-linked Ub chains[15] to support a variety of cellular functions, such as protein trafficking, signalling regulation or lysosomal degradation. By virtue of such multifaceted activity, the NEDD4 family of HECT ligases is known to contribute to a wide range of physiological and pathologic processes, including immune regulation, viral infection, tumorigenesis and neurological disorders[16].

In this study, we investigate the molecular, cellular and organismal functions of a previously uncharacterised member of the NEDD4 family in *Drosophila* to reveal an unexpected and pivotal role for non-degradative ubiquitination in maintaining the liquid state of germ- and neuronal RNP granules.

## Results

**CG42797/Hecw is the *Drosophila* ortholog of HECW1 and HECW2.** By searching the *Drosophila* genome, we identified a single ortholog of human HECW1 and 2, encoded by the uncharacterised gene CG42797 (https://flybase.org/reports/FBgn0261931.html), whose protein product shares co-linearity and 40% overall similarity with the corresponding human proteins. Extensive amino acid sequence identity is present in two WW domains required for substrate interaction and in the catalytic HECT domain (Fig. 1a and Supplementary Fig. 1a). Similar to the *C. elegans* Ce01588 (https://wormbase.org/species/c_elegans/gene/WBGene00009738#0-9f-10), the *Drosophila* protein lacks the C2 lipid binding domain, which is characteristic of the NEDD4 family, suggesting that CG42797 may be unable to bind membranes[17].

To confirm that CG42797 is a catalytically active HECT ligase, we performed an in vitro self-ubiquitination assay. We found that CG42797 is able to ubiquitinate itself and to generate free polyubiquitin chains, whereas a mutation in the evolutionary conserved catalytic cysteine abrogates both activities (Supplementary Fig. 1a, b). To identify the type of Ub linkage generated by CG42797, we set up an in vitro reaction with K/R mutated Ub molecules and an immunofluorescence experiment with specific antibodies. Results indicate that CG42797, like HECW1 [ref. [18]] and other NEDD4 family members[19], preferentially generates K63-specific Ub chains both in vivo and in vitro (Supplementary

Fig. 1c, d). Based on the functional and structural similarities with its human orthologs, we named the *Drosophila* CG42797 gene 'Hecw'.

Previous studies indicate that human HECW1 and HECW2 are preferentially expressed in neuronal tissue[20,21]. Analysis of various *Drosophila* organs by qPCR and immunoblotting, revealed a preferential expression of *Hecw* in the fly gonads and central nervous system (Fig. 1b, c and Supplementary Fig. 1e, f). In fly ovaries, Hecw displays a broad distribution in the cytoplasm of both somatic and germline tissues (Fig. 1d). Similarly, the adult brain showed cytoplasmic staining exclusively in elav-positive neuronal cells (Fig. 1e). Interestingly, as previously reported for several components of the Ub proteasome system and autophagy pathway controlling protein homeostasis[22,23], the level of Hecw expression in fly heads decreases with aging (Fig. 1c, e and Supplementary Fig. 1g).

**Loss of Hecw causes age-dependent neuronal degeneration.** To study the consequences of loss of *Hecw*, we generated catalytic inactive (CI) mutants and knock out (KO) flies using the CRISPR/Cas9 system. As all the tested mutants and KO lines exhibit identical defects (Supplementary Fig. 2), we hereby describe *Hecw^{CI}* and *Hecw^{KO}* as representative examples.

Both *Hecw^{CI}* and *Hecw^{KO}* homozygous (mutant from now on) flies are viable and do not show macroscopic morphological defects, indicating that *Hecw* is a non-essential gene. Prompted by the reduced expression of Hecw in adult flies, we investigated the behaviour of the mutant flies during aging. To minimise the influence of genetic background, environment, nutrition and mating conditions, we performed a lifespan assay with mixed-sex groups in standard cornmeal food. *Hecw^{CI}* and *Hecw^{KO}* mutant animals displayed reduced longevity with respect to isogenic control lines with a 22% and 24% decrease in median survival, respectively (Fig. 2a, b). Mutant lifespan reduction was even higher at 29 °C (28% median survival reduction, Supplementary Fig. 3a).

*Drosophila* display an aging-related decline in climbing whose anticipation is considered a hallmark of neuronal dysfunction. Remarkably, in climbing assays, both Hecw mutant flies displayed motor function impairment, which was already visible in 20-day-old flies and became extremely severe as the flies aged (Fig. 2c). Again, the phenotype was exacerbated at 29 °C (Supplementary Fig. 3b). Next, we analysed frontal brain paraffin sections and observed that mutants presented extended tissue vacuolisation (Fig. 2d, e), which resulted from neuronal death in the CNS (Fig. 2f). The size and number of vacuoles are significantly higher in mutants compared to isogenic control flies (Fig. 2e) and tissue vacuolisation progressively increases with fly age (Supplementary Fig. 3c).

Importantly, defective neuronal function and morphology are rescued by the reintroduction of a wild-type copy of *Hecw* in homozygous *Hecw^{KO}* animals (Fig. 2d, e and g). Moreover, overexpression of Hecw with the pan-neuronal driver *elav-GAL4* causes reduction in longevity, similarly to that observed in *Hecw* mutants (Supplementary Fig. 3d–f), indicating that Hecw activity needs to be tightly regulated to protect neurons from premature neurodegeneration.

**Loss of Hecw leads to defective oogenesis.** The high expression level of Hecw in fly gonads prompted us to assess gamete formation in mutant flies (Fig. 1 and Supplementary Fig. 1f). In females, egg-laying varies with age. After a peak at day 4 post eclosion, there is a physiological decline in egg production, which is generally reduced by 50% at 40 days. When compared to matched control flies, 20-day-old *Hecw* mutant females lay a

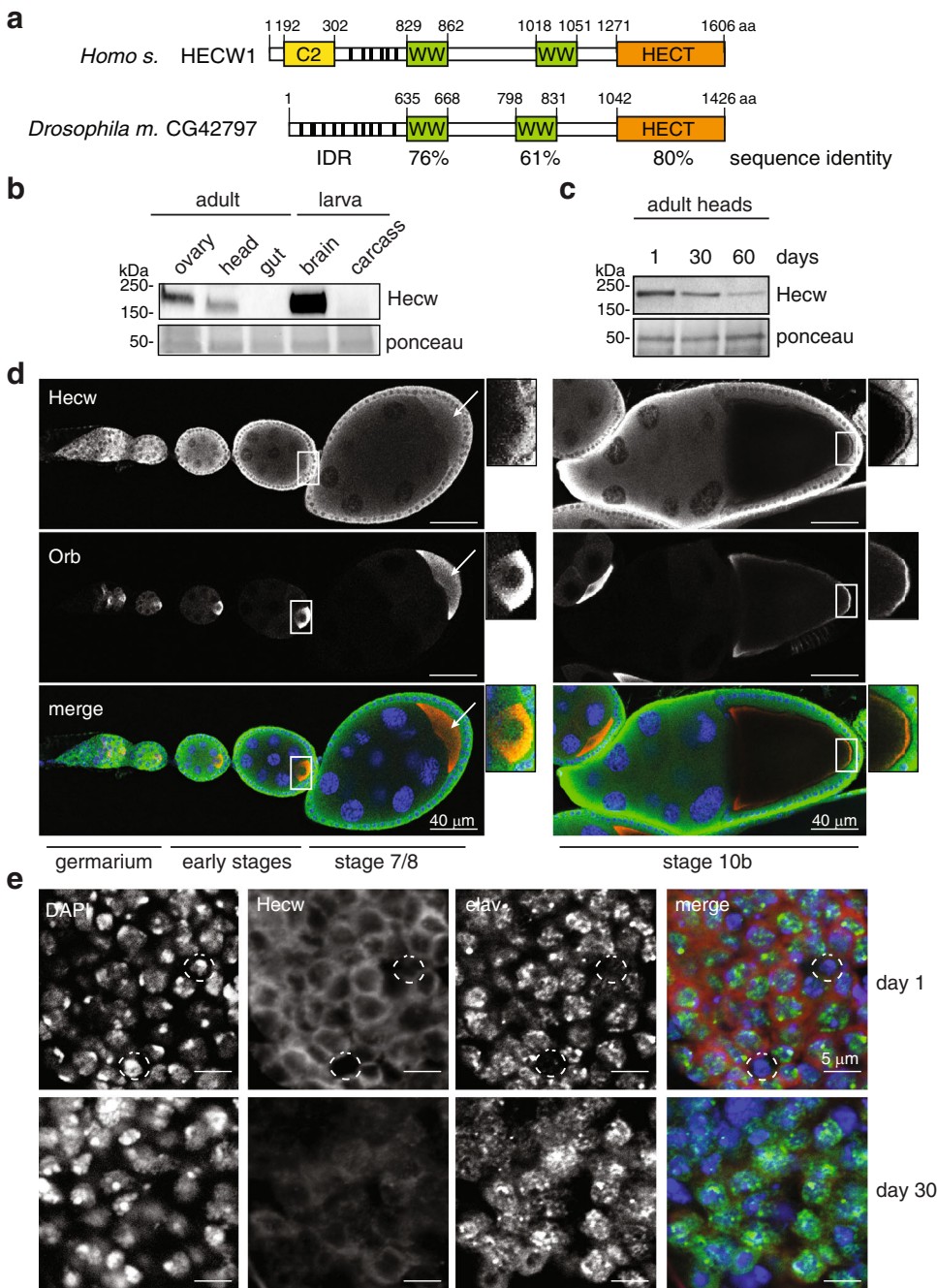

**Fig. 1 Hecw is preferentially expressed in fly gonads and central nervous system. a** Schematic representation of human HECW1 and *Drosophila* CG42797 proteins. Domains shown are: C2 (Ca$^{2+}$ dependent lipid binding) domain in yellow, WW (substrate interacting) domains in green, and HECT (catalytic) domain in orange. The fly ortholog shows no C2 domain and presents an extended intrinsically disordered region (IDR), identified by IUPred (https://iupred2a.elte.hu/). Percentage of identity is reported below the domains. **b** Immunoblot (IB) analysis of the indicated *Drosophila* tissues performed with the anti-Hecw antibody. Ponceau shows equal loading. (n = 3). **c** IB of fly heads at 1, 30 and 60 days with the indicated antibodies. (n = 3). **d** Immunofluorescence (IF) analysis of ovarioles of 3-day-old flies performed with the indicated antibodies. Left panels: germarium and previtellogenic stages 1–7. Right panels: stage 10 egg chamber. Hecw colocalising with Orb (oocyte marker) at the posterior margin of the fly oocyte is indicated by white arrows and highlighted in the magnifications. Scale bar: 40 μm. (n = 3). **e** IF analysis of adult brains of 1-day-(upper panels) or 30-day-old flies (bottom panels) with the indicated antibodies. Neurons are marked with the anti-elav antibody and non-neuronal cells are circled in white. Scale bar: 5 μm. (n = 3).

significantly reduced number of eggs (Fig. 3a). To determine whether the reduced egg-laying exhibited by *Hecw* mutants is accompanied by structural defects, we immunostained ovaries of 3- and 30 day-old well-fed mated wild-type and mutant flies to detect the CPEB protein Orb (oo18 RNA binding), which accumulates in the cytoplasm of the developing oocyte[24]. Already at day 3, 20% of egg chambers derived from mutant flies showed an

altered number of germ cells (Fig. 3b). Defective egg chambers present either reduced or increased number of nurse cells and ring canals, and compound egg chambers (Fig. 3b and Supplementary Fig. 4a). In addition, we detected the presence of late-stage apoptotic egg chambers that are usually absent in nutrient-rich conditions (Fig. 3b), indicating that defective egg chambers are likely culled. Consistently, the hatching rate of mutants is not

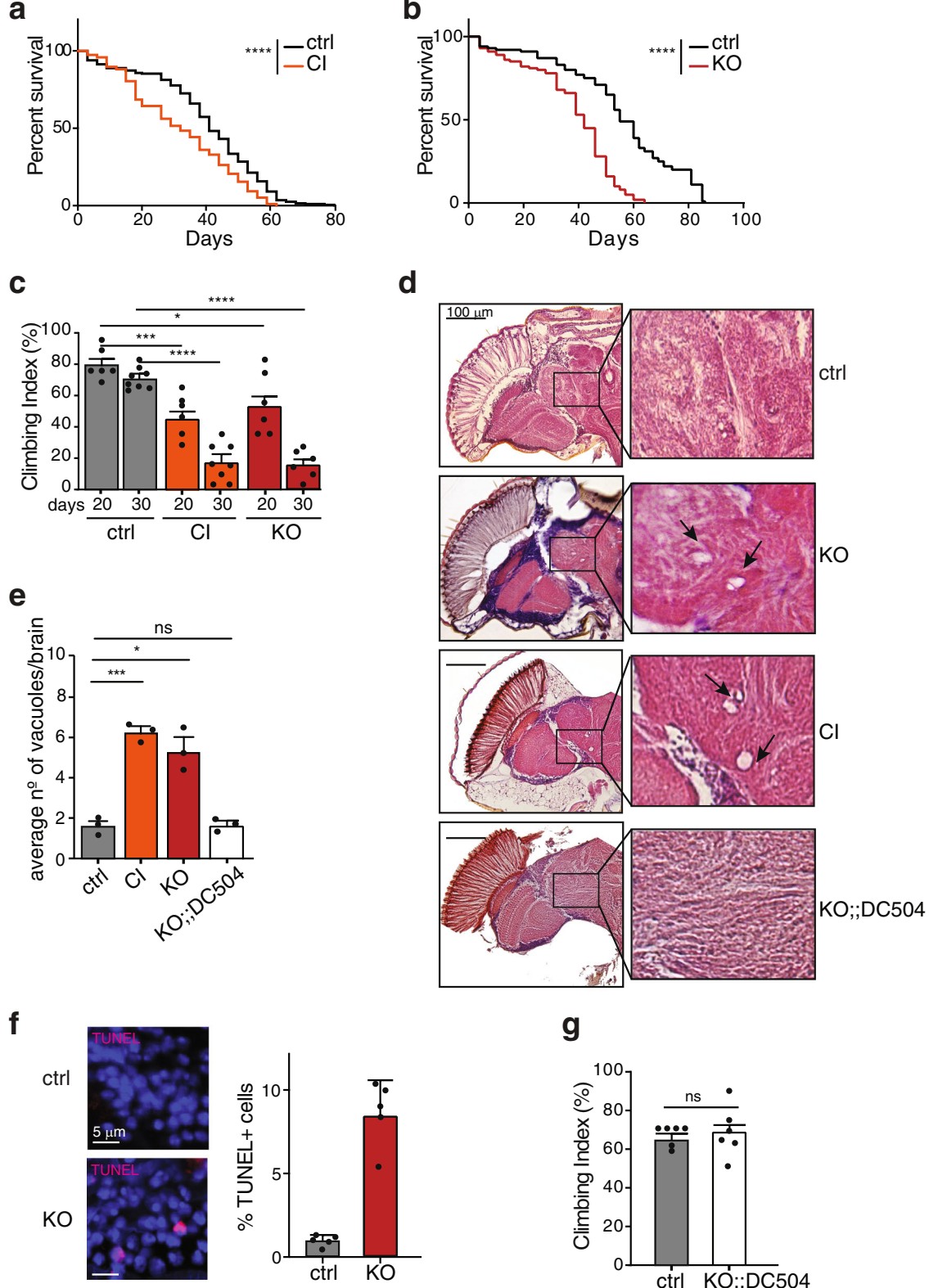

significantly affected (Supplementary Fig. 4b). The penetrance of these defects increases with age, reaching about 40% in 30-day-old flies. Classification and quantification of the aberrant phenotypes observed are summarised in Supplementary Table 1.

To attribute these functional alterations either to the somatic or germline cells (GC), we performed tissue-specific knock down in follicle cells (FC), using the *traffic-jam-Gal4* driver (tj>), and in

GC, using the *nanos-Gal4* driver (nos>). Depletion of Hecw in GC (Fig. 3c) but not in FC (Supplementary Fig. 4c) recapitulates *Hecw*$^{CI}$ or *Hecw*$^{KO}$ mutant phenotypes, demonstrating that aberrant oogenesis defects are mostly caused by the absence of Hecw function in the germline.

The oogenesis defects of chronically Hecw-depleted flies are fully rescued by the reintroduction of a wild-type copy of *Hecw*

**Fig. 2 Hecw mutants display neurodegenerative phenotype. a, b** Survival curve of the indicated genotypes. Percentage of survivals was calculated over 200 animals/genotype. $Hecw^{CI}$ (CI) (**a**) and $Hecw^{KO}$ (KO) flies (**b**) show a significant decrease in their lifespan compared with their control lines. ****$P < 0.0001$ by log-rank (Mantel–Cox) test. **c** Motor function ability measured at the indicated time points by negative geotaxis assay. Results are expressed as climbing index mean ± s.e.m. ($n = 120$ animals/8 independent groups at 30 days, $n = 90$ animals/6 groups for the other time points), *$P < 0.0273$, ***$P < 0.0009$, ****$P < 0.0001$ by unpaired two-tailed $t$-test, comparing the indicated mutant with wild-type. **d** Frontal sections of 30-day-old fly brains of the indicated genotypes stained with H&E and examined by bright-field microscopy. Scale bar, 100 µm. Right, magnification of the central brain regions is highlighted. **e** Quantification of the vacuoles with diameter >2 µm in 30-day-old fly brains of the indicated genotypes. Results are expressed as mean ± s.e.m. of three biological replicates ($n = 5$ animals/genotype/replicate), ***$P = 0.0003$, *$P = 0.0167$, ns by unpaired two-tailed $t$-test. The rescue line HecwKO;;DC504/+ was generated by crossing Hecw mutant flies with Dp(1;3)DC504 animals which contain a duplication of $CG42797$ locus on the third chromosome to analyse animals that are homozygous for HecwKO and heterozygous for Dp(1;3)DC504. **f** TUNEL staining of 30-day-old fly brains of the indicated genotypes. Right, quantification expressed as percentage of positive cells. 200–500 cells were counted for each fly ($n = 5$ animals/genotype). Results are expressed as mean ± SD ****$P < 0.0001$ by two-tailed $t$-test. **g** Negative geotaxis assay performed as in (**c**), on 20-day-old flies of the indicated genotypes. Results are expressed as climbing index mean ± s.e.m. ($n = 48$ animals/6 independent groups). Ns, not significant by unpaired two-tailed $t$-test.

expressed at physiological levels in the whole organism (Supplementary Table 1 and Fig. 3d). As shown in neurons, unscheduled expression of Hecw is also detrimental in gonads, in which ectopic expression of Hecw leads to oogenesis aberrations similar to those observed in $Hecw^{CI}$ animals (Fig. 3e and Supplementary Fig. 4d). These results indicate that tight regulation of Hecw expression in the germline is necessary to allow oocyte development.

**Loss of Hecw changes the state of RNPs in germ cells.** In developing egg chambers, Orb protein expression is usually confined to the oocyte thanks to the repressive action of Cup and Fmrp on selected mRNAs, as part of RNPs assembled in nurse cells[2,25]. Interestingly, we observed the presence of supernumerary Orb-positive cells in about 30% of $Hecw^{CI}$ mutants (Fig. 3b), as well as discrete Orb-positive puncta in the cytoplasm of nurse cells in both $Hecw^{CI}$ and $Hecw^{KO}$ mutant flies (Fig. 4a). To characterise the origin of Orb-positive puncta, we generated $Hecw^{KO}$ mutant flies expressing the RNP marker Me31B::GFP [ref. [26]] in homozygosis. In wild-type flies, association of Orb with RNPs selectively occurs in the oocyte[27,28]. By contrast, in Hecw mutant animals altered Orb puncta fully colocalise with Me31B in the cytoplasm of nurse cells of previtellogenic and early vitellogenic egg chambers (Fig. 4b). Importantly, $Hecw^{KO}$ and $Hecw^{CI}$ egg chambers contain significantly larger Me31B::GFP RNPs when compared to control (Fig. 4b, c and Supplementary Fig. 5a).

The dynamic composition of germline RNPs is linked to the biophysical status of membrane-less organelles, which allows free diffusion in and out of the particles[29,30]. To investigate the nature of RNPs dynamics in the presence or absence of Hecw during *Drosophila* oogenesis, we performed live imaging of $Me31B::GFP$, $Hecw^{CI};Me31B::GFP$ or $Hecw^{KO};Me31B::GFP$ egg chambers (Supplementary Movie 1 and 2). Wild-type RNP particles appear mostly spherical and occasionally undergo fusion (Supplementary Movie 3 and Supplementary Fig. 5b), two features of organelles with liquid-like properties[31]. This behaviour is altered in $Hecw^{KO}$ RNPs, which grow more irregularly shaped aggregates during the coarsening process (Supplementary Movie 2 and Supplementary Fig. 5c).

To determine the nature of the interactions within RNP particles, egg chambers were treated with 2,5% 1,6 hexanediol, an aliphatic alcohol that disrupts weak hydrophobic bonds, typical promoters of liquid droplets[32]. While the size and number of wild-type granules is drastically reduced upon alcohol treatment, the larger $Hecw^{CI}$ and $Hecw^{KO}$ RNPs barely dissolve, mirroring their solid-like nature (Fig. 4d, e).

This result prompted us to evaluate the movement of Me31B particles, which is a combination of free diffusion and active, microtubule-dependent transport[33]. Live observation revealed no

statistical differences in granule speed between control and $Hecw^{KO}$ flies. (Supplementary Movie 1, 2 and Supplementary Fig. 5d). Consistently, immunofluorescence analysis of the microtubule cytoskeleton structure in $Hecw^{KO}$ egg chambers showed no major alteration when compared to control (Supplementary Fig. 5e). RNP mobility in each egg chamber was quantified with Differential Dynamic Microscopy (DDM) analysis[34] that provides the average mean square displacement $MSD_0(\Delta t)$ of the granules, from which we extract an effective diffusion coefficient (Fig. 5a–c). Results indicate that RNP transport is not impaired in the absence of Hecw (Fig. 5d). Moreover, the dynamics of Me31B cytoplasmic fraction were not altered as measured by fluorescence recovery after photobleaching (FRAP) experiments (Supplementary Fig. 5f, g).

To measure the exchange rate of Me31B::GFP molecules inside the RNP particles, we performed fluorescence loss in photobleaching (FLIP) measurements (Supplementary Movie 4–7), as the fast movement of the granules precluded direct FRAP analysis. Initial analysis performed considering the neighbouring regions of the bleached area showed that the mobile fraction of Me31B::GFP is reduced in $Hecw^{KO}$ egg chambers (Supplementary Fig. 6a). As the difference cannot be unambiguously attributed to a different rate of exchange in the granules, we refined our analysis considering smaller regions of interest (ROIs) that selectively include either RNPs or cytoplasm (Fig. 6a). No difference in bleaching efficiency is observed on the directly irradiated area (Fig. 6b) while examination of single RNPs showed a significant difference in fluorescence decay time between wild-type and $Hecw^{KO}$ flies ($\tau_{WT} = (1.3 \pm 0.1) \cdot 10^2\ s$ vs $\tau_{KO} = (4.2 \pm 0.1) \cdot 10^2\ s$. $P < 10^{-6}$ (Fig. 6c). Since we matched control and $Hecw^{KO}$ egg chambers for size or spatial distribution of RNPs (Supplementary Fig. 6b–d), this behaviour could only be attributed to a reduction in the exchange rate of the fluorescent protein between the RNP interior and the cytoplasm. This alteration is limited to the granules as the movement of cytoplasmic Me31B::GFP is comparable between wild-type and $Hecw^{KO}$ flies (Fig. 6d). Thus, in the absence of Hecw, RNP particles may undergo a transition to a less fluid, gel-like state.

Overall, these results indicate that Me31B::GFP-positive RNPs possess liquid droplet properties that are regulated by Hecw-mediated ubiquitination.

**Hecw interacts and colocalises with RNPs components.** The altered physical properties of Me31B::GFP RNPs prompted us to investigate whether Hecw may interact with RNP proteins. Consistently, in ovarian extract, Hecw co-immunoprecipitates with Me31B::GFP together with other known RNP components, such as Orb and Fmrp [ref. [2]] (Supplementary Fig. 7a). The interaction is maintained upon RNAse treatment, suggesting that

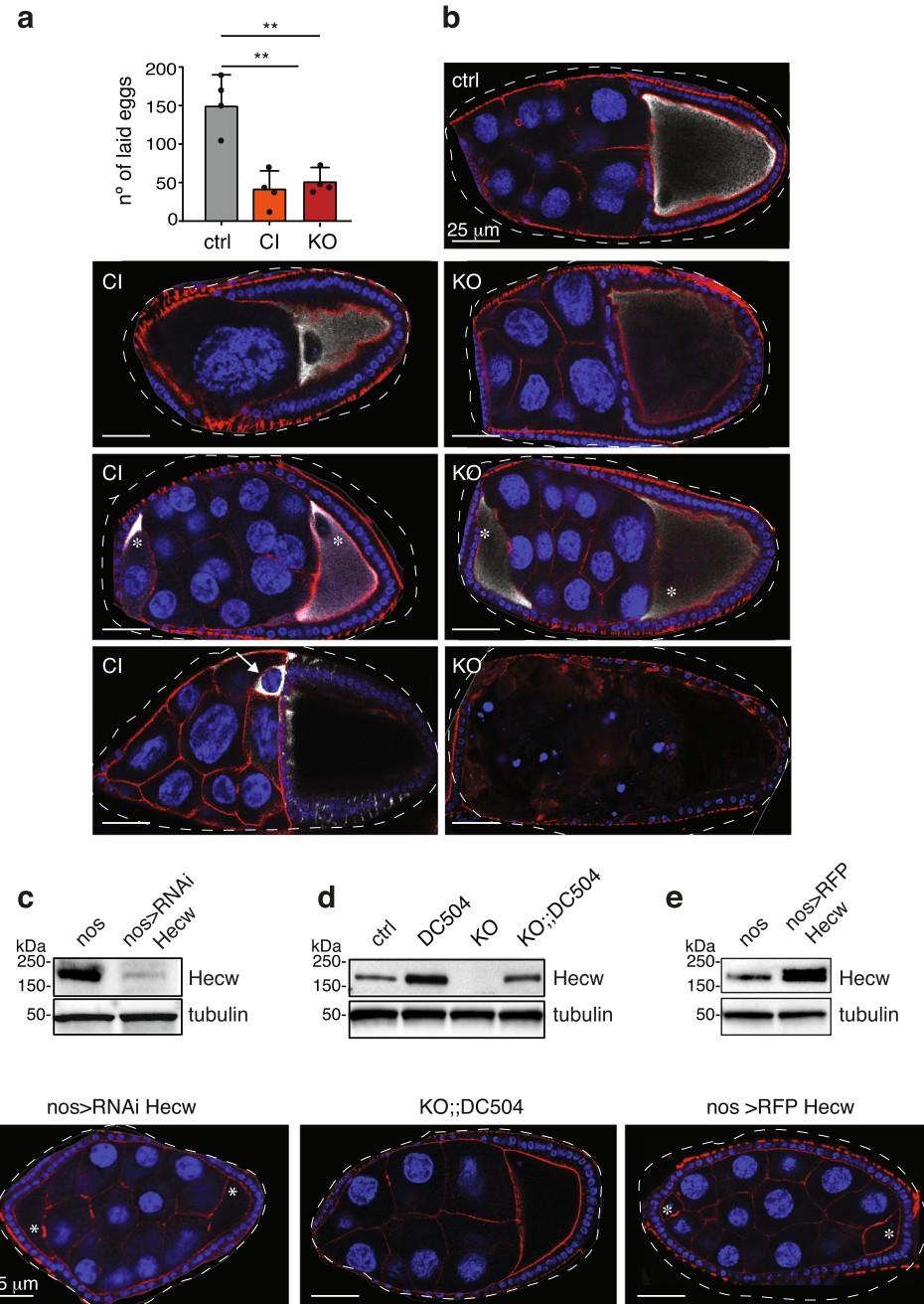

**Fig. 3 Aberrant expression of Hecw impairs proper oogenesis. a** Fertility assay of 20-day-old females of the indicated genotypes. Results of four biological replicates are expressed as mean ± SD, **$P < 0.01$ by unpaired two-tailed $t$-test ($n = 6$ females/genotype/replicate). **b** IF analysis of wild-type (ctrl, upper right panel), $Hecw^{CI}$ (CI, left panels) and $Hecw^{KO}$ (KO, right panels) egg chambers. Mutants show aberrant number of nurse cells (top panels), compound egg chambers (middle panels), oocyte misspecification and apoptotic egg chambers (bottom panels). Blue, DAPI; red, phalloidin; white, Orb; asterisks indicate oocytes in compound egg chambers; white arrow indicates an Orb-positive nurse cell. Scale bar: 25 μm. **c–e** IB and IF analysis of the indicated genotypes. Hecw depletion in the germline causes oogenesis defects, as in (**b**), upper panel. Example of compound egg chamber in a 3-day-old Hecw-depleted fly ($nos$>$RNAi$ $Hecw$) (**c**). The oogenesis defects are rescued in the $Hecw^{KO:DC504}$ genetic background (**d**). Ectopic expression of Hecw in the germline ($nos$>$RFP$-$Hecw$) induces oogenesis defects (**e**). Blue, DAPI; red, phalloidin. Asterisks indicate oocytes of the compound egg chamber. Scale bar: 25 μm. See also Supplementary Table 1 for the quantification.

Hecw interacts directly with RNPs via protein-protein interactions (Fig. 7a).

To identify Hecw binding partners, we performed a pull-down assay using a GST construct spanning the two WW domains as a bait, followed by mass spectrometry (Supplementary Table 2). The majority of Hecw interactors are mRNA binding proteins involved in multiple steps of RNA processing (Supplementary Table 2), among them Fmrp. Strikingly, two other interactors, Hrp48 and

Glorund, are translational repressors previously implicated in translational control during oogenesis[35]. These findings provide compelling evidence for a critical role of Hecw in RNPs regulation.

**Hecw-dependent ubiquitination modulates Orb via Fmrp.** How could Hecw-dependent ubiquitination regulate RNPs? We focused on Fmrp, a known RNA binding repressor of the Orb autoregulatory loop in nurse cell[25]. $Fmr1^{\Delta113}$ loss of function

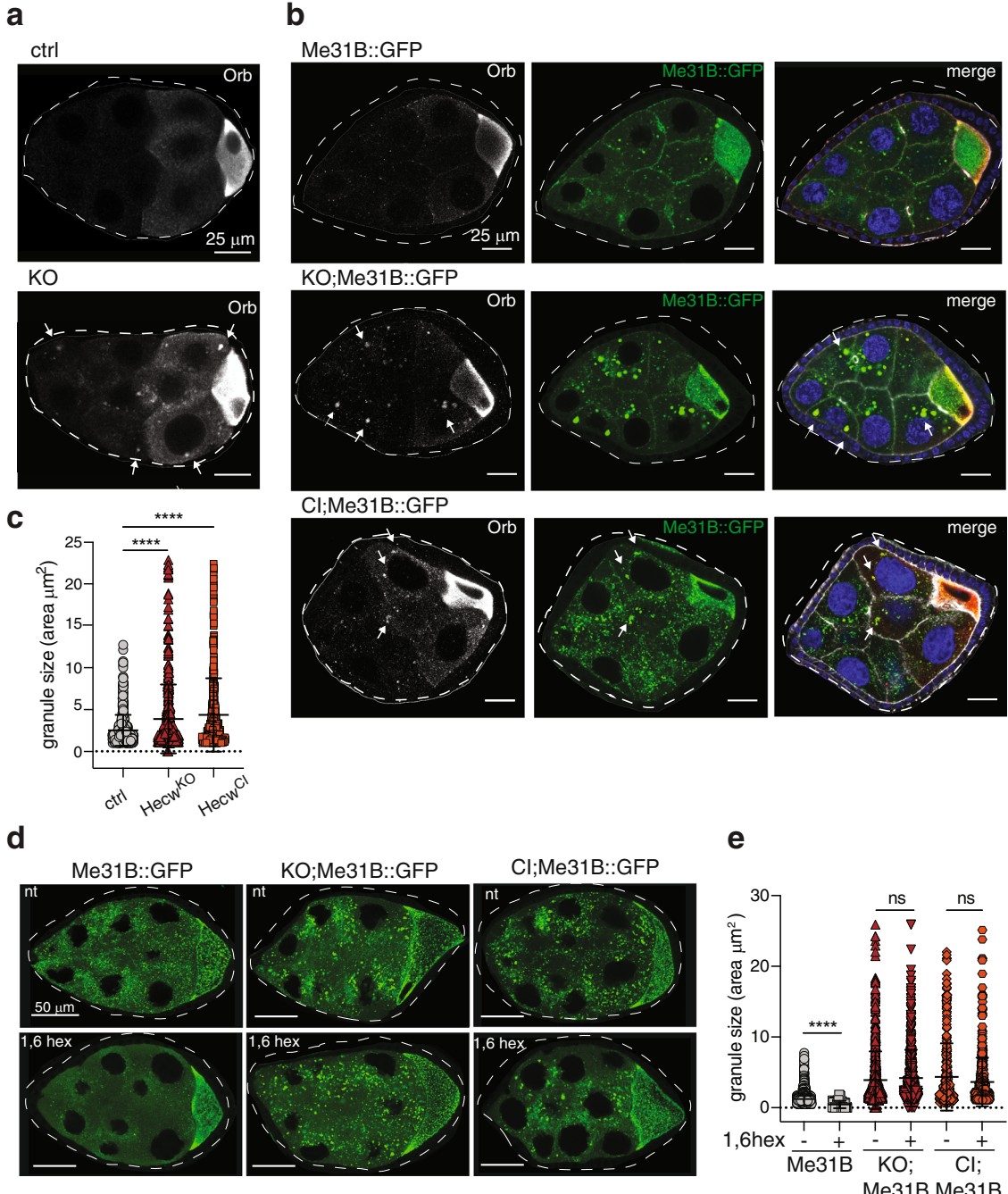

**Fig. 4 RNP morphology and composition is altered in Hecw$^{KO}$. a** IF analysis of egg chambers from 3-day-old flies of the indicated genotypes with anti-Orb antibody ($n = 3$). White arrows indicate ectopic Orb puncta in nurse cells of *Hecw$^{KO}$* flies. **b** IF analysis of the indicated genotypes ($n = 3$). Green, Me31B::GFP which marks the RNPs; red, Orb which marks the oocyte (in white in the first panel); blue, DAPI; white, phalloidin. Scale bar: 25 µm. **c** Quantification of RNPs size of egg chambers (stages from 6–9) from 3-day-old flies of the indicated genotypes. Results are expressed as mean ± SD. $n = 477$ (ctrl Me31B::GFP), $n = 559$ (Hecw$^{KO}$;;Me31B::GFP), $n = 444$ (Hecw$^{CI}$;;Me31B::GFP), 5 egg chambers/genotype from two biological replicates. ****$P < 0,0001$ by two-tailed Mann–Whitney test. **d** IF analysis of egg chambers of 3-day-old flies as indicated ($n = 3$), not treated (nt) or treated with 1,6 hexanediol (1,6 hex). Scale bar: 50 µm. **e** Quantification of RNPs size in egg chambers (stage 7–9) of the indicated genotypes and conditions. Treatment significantly reduces RNP size in control but not in *Hecw$^{KO}$* and *Hecw$^{CI}$* flies (KO;Me31B::GFP and CI;Me31B::GFP). Particles analysed: $n = 505$ ctrl (nt), $n = 49$ ctrl (1,6 hex), $n = 642$ Hecw$^{KO}$ (nt), $n = 536$ Hecw$^{KO}$ (1,6 hex), $n = 314$ Hecw$^{CI}$ (nt), $n = 553$ *Hecw$^{CI}$* (1,6 hex) (3 egg chambers/genotype). Size mean ± SD is reported. ****$P < 0.0001$ by two-tailed Mann–Whitney test.

mutant flies[25] display oogenesis defects that resemble the ones of *Hecw* mutants (Fig. 7b). Corroborating the idea that the two proteins act on the same axis, genetic interaction revealed no worsening of the phenotypes (Supplementary Fig. 7b). In addition, we found that endogenous Hecw and Fmrp from fly ovaries

coimmunoprecipitate (Fig. 7c) and that Fmrp is a substrate of Hecw, as measured by in vitro ubiquitination assay (Fig. 7d). We confirmed this result in vivo by isolating endogenous ubiquitinated targets using tandem-repeated Ub-binding entities (TUBEs), a sensitive method that demonstrates the covalent

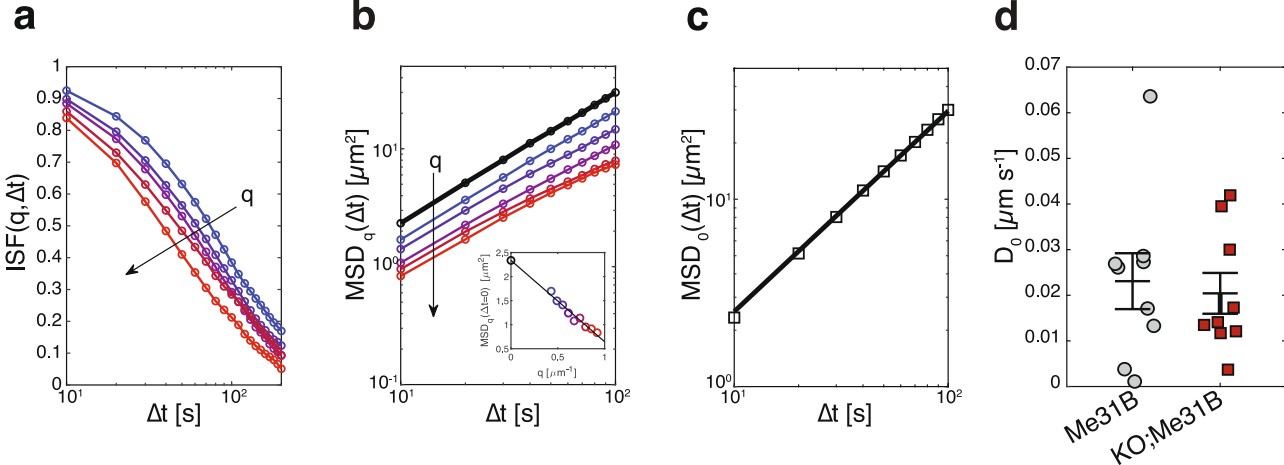

**Fig. 5 Lack of Hecw does not alter RNPs mobility. a** Representative intermediate scattering functions $f(q, \Delta t)$ obtained from Differential Dynamic Microscopy (DDM) analysis of an image sequence of a KO;Me31B::GFP egg chamber plotted as a function of the time delay $\Delta t$ for different values of the wave vector $q$ in the range $0.4-0.8\,\mu m^{-1}$. **b** From each of the curves shown in (**a**), a $q$-dependent estimate of the mean square displacement $MSD_q(\Delta t) \equiv -\frac{4}{q^2}\ln f(q, \Delta t)$ can be obtained (coloured symbols). Black circles correspond to the best estimate for the average mean square displacement $MSD_0(\Delta t)$ of the RNPs, obtained by extrapolating $MSD_q(\Delta t)$ to $q = 0$ at fixed $\Delta t$. The extrapolation procedure is illustrated in the inset for the representative case $\Delta t = 10s$. **c** Average mean square displacement $MSD_0(\Delta t)$ (symbols) of the RNPs in the considered egg chamber, displaying a characteristic linear dependence on the delay time $\Delta t$, which is indicative of a random, diffusive-like motion. An effective diffusion coefficient $D_0$ can be obtained by fitting a linear model $MSD_0(\Delta t) = 4D_0\Delta t$ to the data (continuous black line). **d** DDM analysis of $N = 18$ egg chambers shows no significant difference in the RNP mobility between wild-type (Me31B, $D_{0,WT} = 0.023 \pm 0.006\ \mu m^2/s\ N = 9$) and $Hecw^{KO}$ egg chambers (KO;Me31B, $D_{0,KO} = 0.021 \pm 0.005\ \mu m^2/s$, $N = 9$). See also Supplementary Fig. 5h, i.

attachment of polyubiquitin chains to proteins modified at low stoichiometry[36]. While Fmrp is clearly detectable in the pull-down from wild-type ovaries, the lane corresponding to the $Hecw^{KO}$ ovaries displays a remarkably reduced amount of protein (Fig. 7e), indicative of reduced Fmrp ubiquitination in the absence of the E3 ligase.

To evaluate the impact of Hecw-dependent ubiquitination on Fmrp, we examined its protein levels in wild-type and $Hecw$ mutant ovaries. As predicted by Hecw specificity towards K63-linked polyubiquitination (Fig. 7d), Fmrp stability is not affected by the lack of Hecw activity (Fig. 7f). In contrast, the downstream target Orb accumulates in $Hecw$ mutant ovaries (Fig. 7f), in good agreement with ectopic presence of Orb puncta in nurse cells (Fig. 4a, b). Regulation by Hecw appears to occur at the protein level, since both $Fmr1$ and $Orb$ mRNA expression does not change significantly when compared with wild-type controls (Supplementary Fig. 7c). Consistent with the antagonistic effect exerted by Hecw on Orb through Fmrp, overexpression of Hecw in the germline strongly downmodulates Orb protein levels (Fig. 7g) without affecting its mRNA (Supplementary Fig. 7d) and treatment of ovaries with the proteasome inhibitor MG132 does not rescue Orb expression (Supplementary Fig. 7e).

Me31B-positive RNPs in ovaries control spatial and temporal translation of key mRNAs during development[26]. To test whether the aberrant RNP structure and composition of $Hecw$ mutant tissue alters translation control, we analysed the localisation of the two major Orb target proteins: Gurken (Grk) and Oskar (Osk). While the mRNA levels are not altered (Supplementary Fig. 7c), both proteins show aberrant localisation in $Hecw$ mutant oocytes. Grk is found ectopically in the ventral portion of the oocyte (Fig. 7h), while Osk localises away from the posterior margin (Fig. 7i). Remarkably, in stage 8/9, ectopic Osk colocalises with Me31B::GFP positive particles in $Hecw$ mutant egg chambers, suggesting premature translation (Supplementary Fig. 7f), as is the case of Orb. To prove that these phenotypes are causally linked to aberrant Orb expression, we reduced Orb levels by crossing both KO and CI $Hecw$ flies with $Orb^{343/+}$ flies (Supplementary Fig. 8a). The results

showed that we could largely rescue the defective egg chambers present in $Hecw$ mutant flies, leaving altered RNPs unaffected (Supplementary Table 1 and Supplementary Fig. 8b, c). Overall, these results indicate that Hecw is a positive regulator of Fmrp repressor activity on Orb and its targets.

**Hecw modulates Fmrp activity in neurons.** Neural synapses, like oocytes, depend heavily on translating stored mRNA[3]. Fmrp exerts a critical role in the presynaptic compartment where it regulates the transport and translation of specific mRNAs, which have an impact on synaptic strength and connectivity[37]. One of the relevant Fmrp neuronal mRNA targets is $Profilin$[38] whose mutations cause familiar ALS[39]. We thus wondered if, also in neurons, the $Hecw$ mutant may modulate Fmrp activity and, as a consequence, the expression of its repressed target. Remarkably, while Fmrp stability is not affected by an altered expression of $Hecw$, Profilin shows a twofold increase in the $Hecw^{KO}$ line and a similar decrease upon Hecw overexpression in neurons, without significantly affecting its mRNA (Fig. 8a). We then monitored RNPs status, revealing enlarged Me31B-positive RNPs in the neurons of young adult $Drosophila\ Hecw^{KO}$ brains (Fig. 8b). In addition, treatment with 2.5% 1,6 hexanediol significantly reduced the size and number of wild-type granules, but had minimal effect on the larger granules present in the $Hecw$ mutant brains (Fig. 8c, d). These findings confirm the critical role of Hecw in maintaining the liquid state of RNP granules and suggest $Profilin$ as one of the pathologically relevant proteins downstream of Hecw activity in neurons.

To prove that Orb and Profilin dysregulation in Hecw mutants depends on the translational repression of Fmrp in the granules, we decided to downmodulate Fmrp by RNAi using $nanos$- and $elav$-Gal4 driver, respectively. Strikingly, immunoblot analysis showed that the levels of both Orb and Profilin in ovaries and neurons, respectively, are restored upon Hecw overexpression when Fmrp levels are strongly reduced (Fig. 8e). Thus, our genetic analysis reveals the existence of a Hecw-Fmrp-Orb/Profilin axis that impacts on RNP translational repression activity.

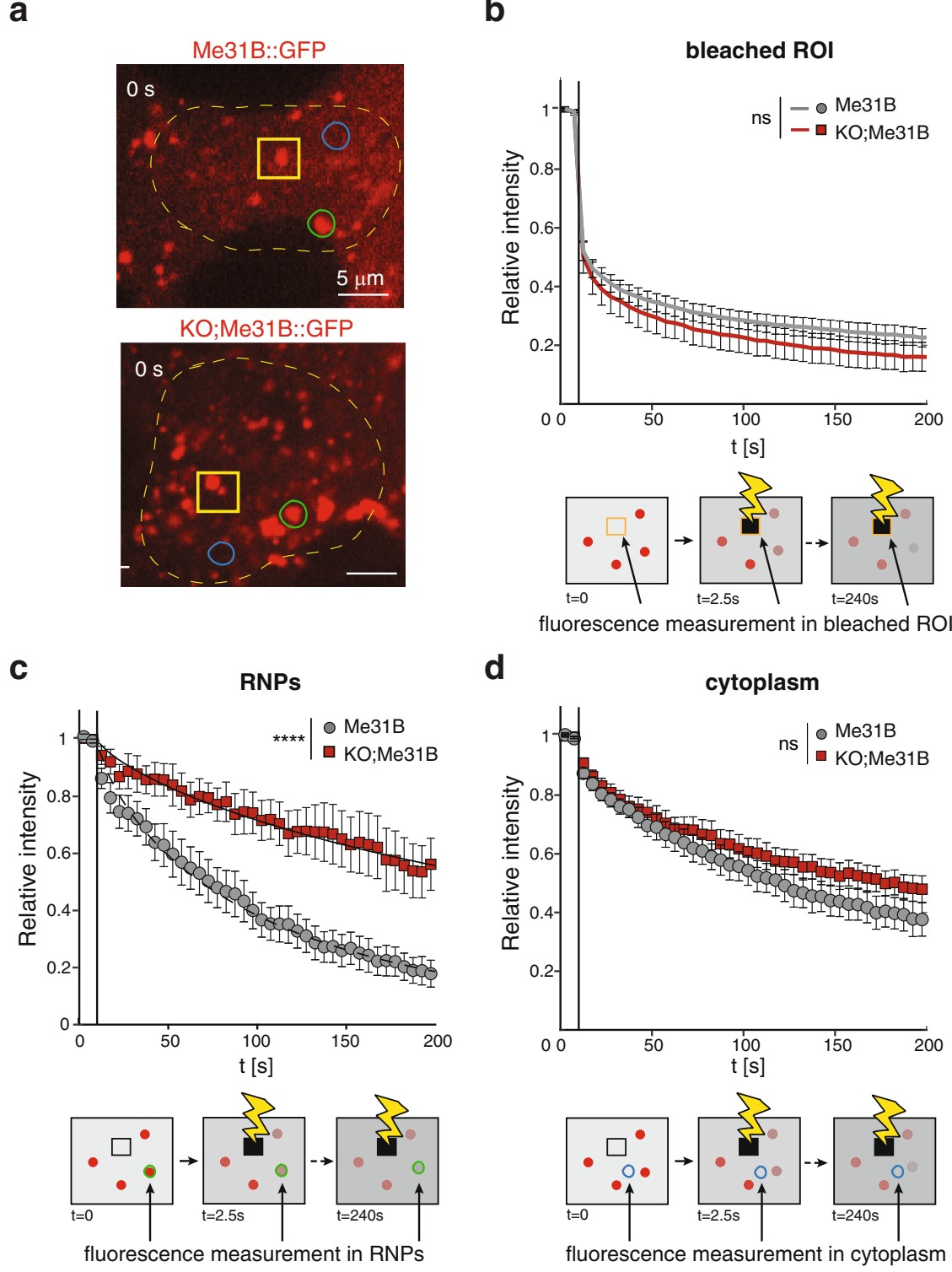

**Fig. 6 Lack of Hecw alters the RNP biophysical proprieties. a** FLIP analysis was performed by repeatedly bleaching a randomly selected ROI (indicated by a yellow square) and by monitoring and quantifying the fluorescence level as a function of time in the same region, in single RNPs outside the bleached area (green circle), and in neighbouring portions of the cytoplasm (blue circle). One representative image for each genotype is shown. In (**b**), (**c**) and (**d**), the relative intensity of Me31B::GFP fluorescence in the three regions: bleached area, RNPs and cytoplasm, respectively, is plotted as a function of time for both control and $Hecw^{KO}$. Data are reported as mean ± SD. ****$P < 0.0001$ by two-tailed Mann–Whitney test ($n = 18$ samples/genotype). Both in the bleached area and in the cytoplasm, the loss curves show no statistically significant difference ($P = 0.07$ by $t$-student) between control and $Hecw^{KO}$, whereas a highly significant difference is observed for RNPs.

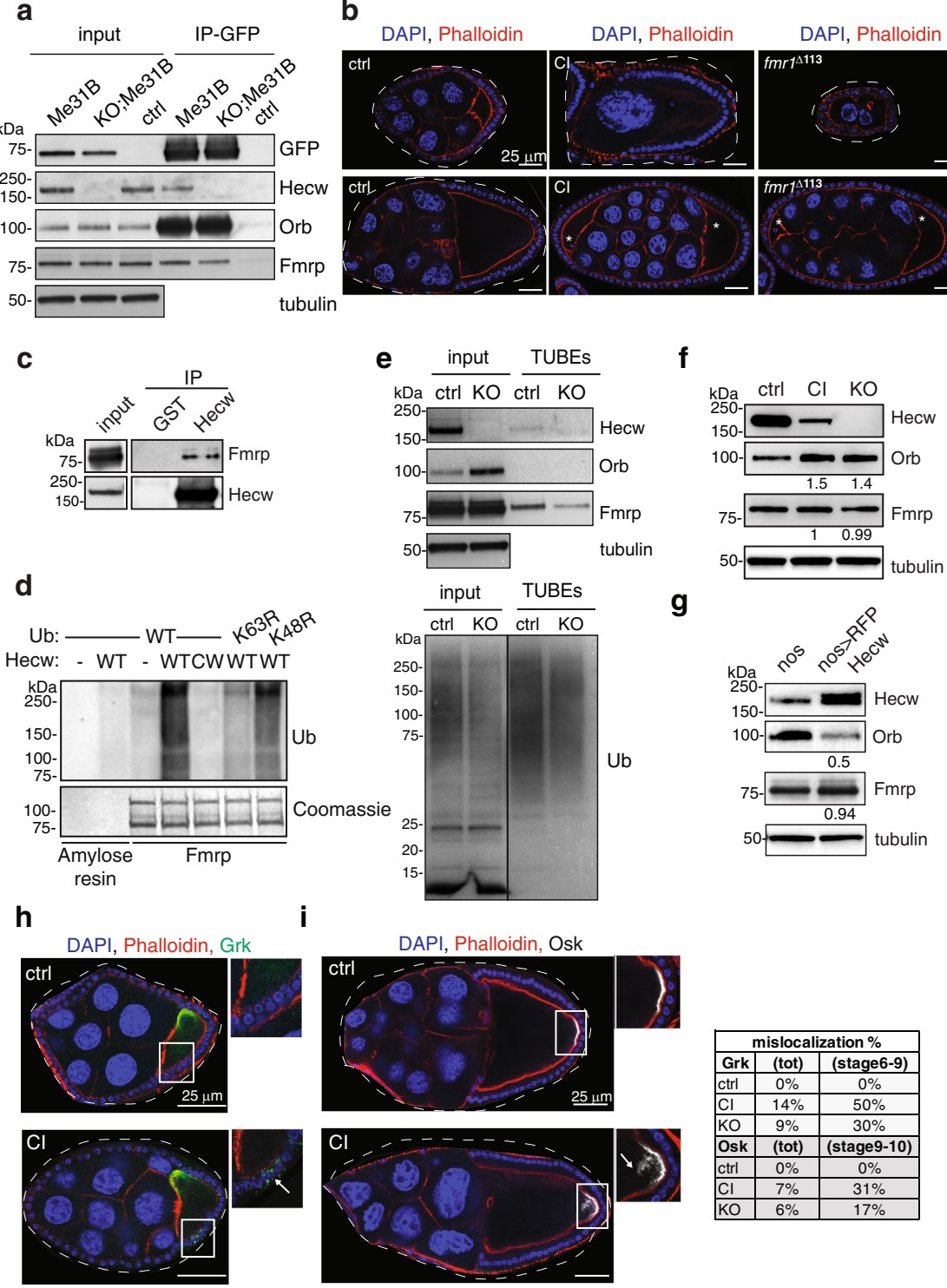

**Ub controls the RNP liquid state and Fmrp repressor activity.** To determine whether ubiquitination may directly affect Fmrp activity in RNP granules, we reconstituted a model biomolecular condensate by combining a purified recombinant fragment of *Drosophila* Fmrp, a lysate of reticulocytes and a luciferase mRNA as a reporter for protein synthesis[40]. Previous studies have shown that the isolated disordered region containing the RGG motif of human FMRP is sufficient for both phase separation and translational repression in vitro[40]. Based on sequence alignment, we generated the corresponding *Drosophila* Fmrp construct as well as a chimeric version with Ub fused in frame at the C-terminus as

a proxy for ubiquitinated Fmrp (Fig. 9a). We then compared the two constructs using differential interference contrast (DIC) and luminescence assay to examine formation of phase-separated protein-rich condensates (Fig. 9b) and to measure translation inhibition (Fig. 9c). Confirming previous reports[40,41], a switch-like translation inhibition response that parallels phase separation is observed increasing the concentration of the RGG construct (Fig. 9b, c). Intriguingly, the RGG-Ub phase separates at a approximate twofold lower concentration shifting the inhibitory curve to the right (Fig. 9b, c). Thus, ubiquitination appears to increase Fmrp phase-separation propensity together with its

**Fig. 7 Hecw interacts with RNP components and ubiquitinates Fmrp. a** 0.5 mg ovary lysates from *Me31B::GFP* (Me31B), *Hecw^KO1^;Me31B::GFP* (KO;Me31B) and control (yw) lines were IP with anti-GFP antibody in the presence of 100 ug/ml RNAseA. Input corresponds to 2% of the total immunoprecipitated proteins. IB as indicated. (n = 2). **b** IF analysis of egg chambers from 3-day-old wild-type (ctrl, left), *Hecw^CI^* (CI, middle) or *Fmr1* mutant (*Fmr1^Δ113^*, right) flies (n = 2). Mutant flies present same defects with similar penetrance. Examples of reduced number of nurse cells (upper panels) and compound egg chambers (bottom panels) are reported. Red, phalloidin; blue, DAPI. Asterisks indicate oocytes of the compound egg chamber. Scale bar: 25 μm. **c** 0.5 mg of ovary lysates from the indicated lines were IP and IB as indicated. Input corresponds to 1% of total immunoprecipitated proteins. (n = 2). **d** In vitro ubiquitination assay of Fmrp with ubiquitin wild-type, K63R or K48R mutants analysed by IB with anti-Ub antibody. Coomassie shows equal loading. (n = 3). **e** 0.6 mg of ovary lysates from the indicated lines were IP for Ub with TUBEs and IB as indicated. Input corresponds to 3% of total immunoprecipitated proteins. (n = 2). **f** IB of wild-type (ctrl), *Hecw^CI^* (CI) and *Hecw^KO^* (KO) fly ovaries with the indicated antibodies. Orb levels are increased upon Hecw depletion. Values reported below each blot are normalised for tubulin and represent the fold change relative to the control, presented as mean of three experiments. **g** IB of control (driver only, *nos-GAL4*, nos) and RFP-Hecw-overexpressing fly ovaries (*nos>RFP Hecw*) with the indicated antibodies. Orb levels are decreased upon Hecw overexpression in the germline. **h, i** IF analysis of egg chambers of 3-day-old wild-type and CI with the indicated antibodies. (n = 2). White arrows in the magnifications indicate mislocalised Grk (green, **h**) at the ventral side of the oocyte or mislocalised Osk (white, **i**) diffused from the posterior margin of the oocyte in the Hecw mutant flies. Scale bar: 25 μm. Percentage of mislocalised proteins (counted overall egg chambers and in specific stages) are indicated in the table on the right.

inhibitory activity in vitro, and, by extension, it may be a critical determinant that facilitates RNPs assembly in ovaries and neurons.

## Discussion

We report here the identification and molecular characterisation of Hecw, the fourth member of the NEDD4 family in *Drosophila*. Flies lacking Hecw activity show neurodegenerative traits, shorter lifespan and reduced fertility. We provide compelling evidence that Hecw-mediated ubiquitination is needed for the diffusive exchange in and out of RNP compartments during oogenesis and in neurons. Mechanistically, Fmrp ubiquitination by Hecw is required for the translational repression of Orb and Profilin (Fig. 9d), two known Fmrp targets[25,38].

**Ubiquitin controls RNPs liquid state.** In the *Drosophila* ovary, sponge bodies are marked by the de-capping activator/RNA helicase Me31B [ref. [26]]. These are first synthesised in the nurse cells and then transported and localised posteriorly in developing oocytes, where their mRNAs are translated, ultimately dictating the establishment of germ cells and embryonic axes[42,43]. While their composition has been recently investigated[4,44,45], the mechanism behind the dynamic nature and biophysical properties of sponge bodies are largely unknown. Recent literature underlined a liquid-like behaviour for RNP granules in *Caenorhabditis elegans*[29,46], as well as for germ granules in fly embryos[47]. We now show that also *Drosophila* sponge bodies behave like liquid droplets.

Our in vivo data provide evidence for the pivotal role of Hecw and Ub in maintaining the liquid-like nature of RNPs, and show that, in the absence of Hecw, they transition into a less dynamic gel-like state. The lower exchange rate between RNPs and the cytoplasm is specific of *Hecw* mutant granules and possibly reflects an altered internal rearrangement and activity of the proteins present inside these membrane-less particles. Indeed, the Orb protein that is undetectable in the nurse cells granules[27,28], is clearly present in the larger Hecw-depleted RNPs, suggesting a lack of local translational repression. Genetic validation demonstrates that in an *Orb* heterozygous background, we could largely rescue the defective phenotype of egg chambers observed in *Hecw^KO^* and *Hecw^CI^* homozygous flies.

Our in vitro reconstitution studies further sustain the idea that Hecw-mediated ubiquitination favors phase-separated transitions, and promotes translation repression of granules generated by recombinant Fmrp. Indeed, the protein concentration required for LLPS is shifted to lower values when Ub is coupled to the RGG domain of Fmrp. These data suggest that Ub acts as a

stabiliser of phase separation, in good accordance with its peculiar biophysical and thermodynamic properties.

An important question that remains to be addressed is the impact of linkage-specific polyubiquitin chains on RNP granules. Longer chains are predicted to increase multivalency further promoting LLPS, in particular in the case of K63-linked chains whose moieties are arranged in a beads-on-string scaffold. Other chains may have alternative effects depending on their different structural conformations and on the nature of the RNPs components, as already demonstrated for UBQLN2, hHR23B, and p62, whose condensates could either assemble (hHR23B, and p62) or disassemble (UBQLN2) via noncovalent interactions with polyubiquitin chains[48].

Although our study is limited to the analysis of Me31B in oocytes and neurons, it is conceivable that other RNPs may behave similarly and that Hecw-mediated ubiquitination may modulate liquification of other phase-separating complexes, thus promoting specific regulated events.

**Hecw acts in the Fmrp-Orb and Fmrp-Profilin axis.** We identify Fmrp as interactor, substrate and effector of Hecw activity in RNPs. Importantly, a Hecw mutant phenocopies Fmrp loss-of-function flies, both in terms of phenotypic defects and molecular targets. Indeed, Orb, a known target of Fmrp repressor activity[25], shows an inverse correlation with Hecw expression, an effect that cannot be ascribed to a direct activity of Hecw on Orb. Same evidence was obtained in neurons where Profilin appears to be a relevant Fmrp target[38]. Thus, similar to Fmrp [ref. [49]], changes in Hecw expression levels are deleterious and induce pathological effects. Based on this evidence, we proposed and genetically validated a linear cascade of events in which Hecw ubiquitinates Fmrp to maintain Orb and Profilin mRNAs in a repressed state. Such K63-linked, non-proteasomal function of Hecw in Fmrp ubiquitination is in good accordance with the previously reported activity of this enzyme family[18,19], and is clearly different from the proteasome-mediated degradation of Fmrp identified in neurons[50].

Fmrp represses translation both during ovary and neuronal development, however, the underlying mechanism remains unclear[51,52]. We envisage that Ub, by adding an additional surface of interaction to Fmrp, could enable precise spatiotemporal recruitment of repressive components[53] and ribosomes[54]. Indeed, while *Drosophila* Fmrp directly binds the 80 S ribosome near the tRNA and translation factor binding site, post-translational modifications (PTMs) have been suggested to modulate the affinity of Fmrp for the ribosome or target mRNAs, thereby "turning off" protein synthesis locally[54]. A Ub-dependent Fmrp association to ribosomes might occur in nurse cells and be

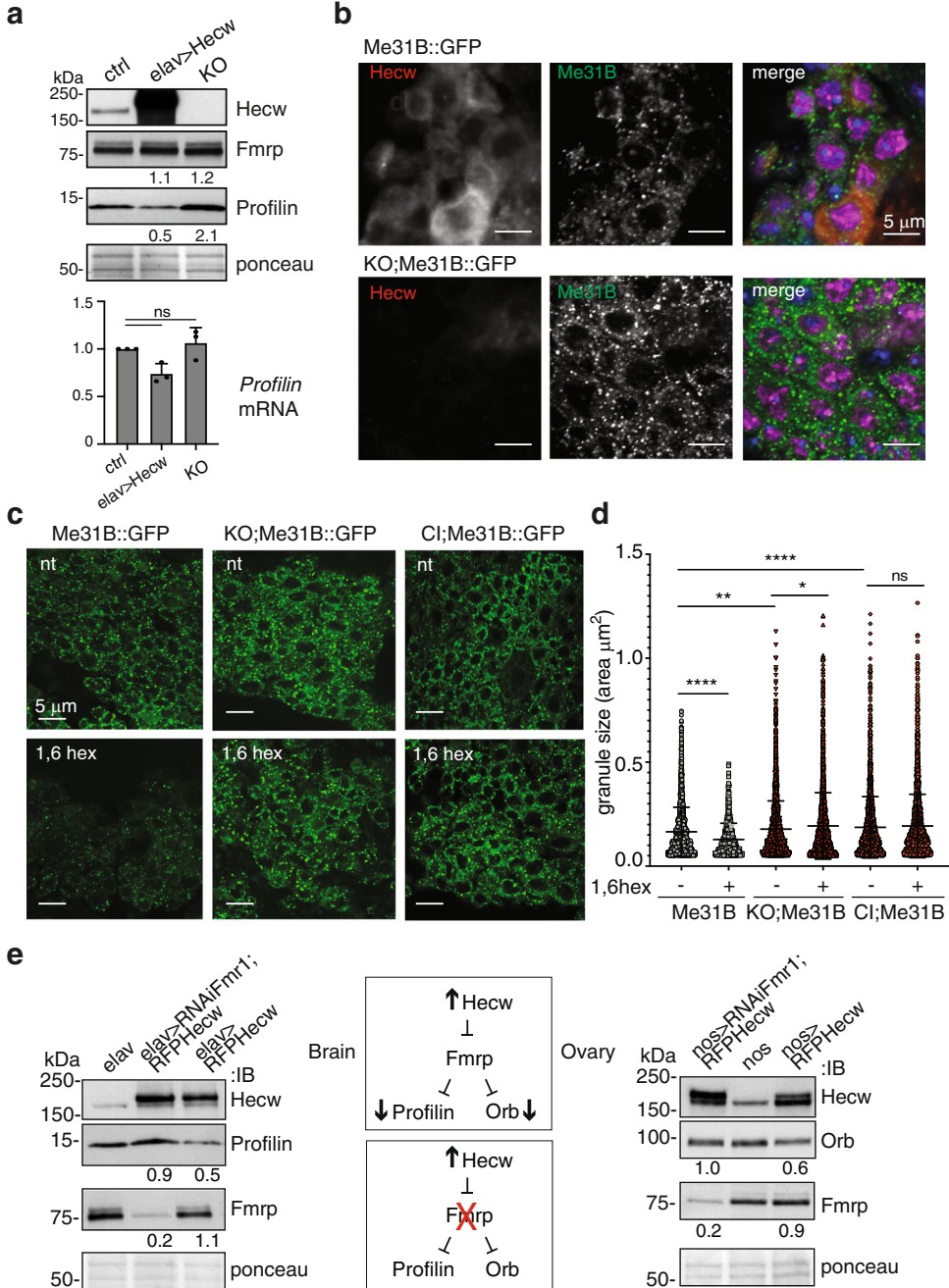

**Fig. 8 Lack of Hecw alters RNP in neurons. a** Upper panel: IB of wild-type (ctrl), Hecw$^{KO}$ (KO) and RFP-Hecw-overexpressing (elav>RFP Hecw) fly heads with the indicated antibodies. Profilin levels increase in Hecw$^{KO}$ and decrease upon Hecw overexpression. Values reported below each blot are normalised for tubulin and represent the fold change relative to the control, presented as mean of three experiments. Bottom panel: mRNA level of profilin was measured by qPCR in the adult heads of the indicated genotypes. The reported expression levels relative to control are expressed as mean ± SD calculated over three experiments. Ns, not significant by unpaired two-tailed t-test. **b** IF analysis of adult brains from 1-day-old flies with the indicated antibodies (n = 2). Neurons are marked with the anti-elav antibody (purple); red, Hecw; blue, DAPI. Scale bar, 5 μm. **c** IF analysis of adult brains from 1-day-old flies of the indicated genotypes, not treated (nt) or treated with 1,6 hexanediol (1,6 hex). (n = 3). Scale bar, 5 μm. **d** Quantification of RNP size in fly neurons of the indicated genotypes and conditions. RNP size is significantly increased in Hecw$^{KO}$ flies (KO;Me31B::GFP **P < 0.01) and Hecw$^{CI}$ (CI;Me31B::GFP **P < 0.01). 1,6 hexanediol treatment significantly reduces RNP size in control flies. Size mean ± SD is reported. ****P < 0.0001 by two-tailed Mann–Whitney test (ten heads/genotype from two biological replicates). **e** Profilin and Orb levels are restored upon concomitant Hecw overexpression and Fmr1 silencing. Left panel: Profilin expression in fly brains of the indicated genotypes. Values reported below each blot are normalised for tubulin and represent the fold change relative to the control, presented as mean of three experiments. Right panel: Orb expression in fly ovaries of the indicated genotypes. Values reported below each blot are normalised for tubulin and represent the fold change relative to the control, presented as mean of two experiments.

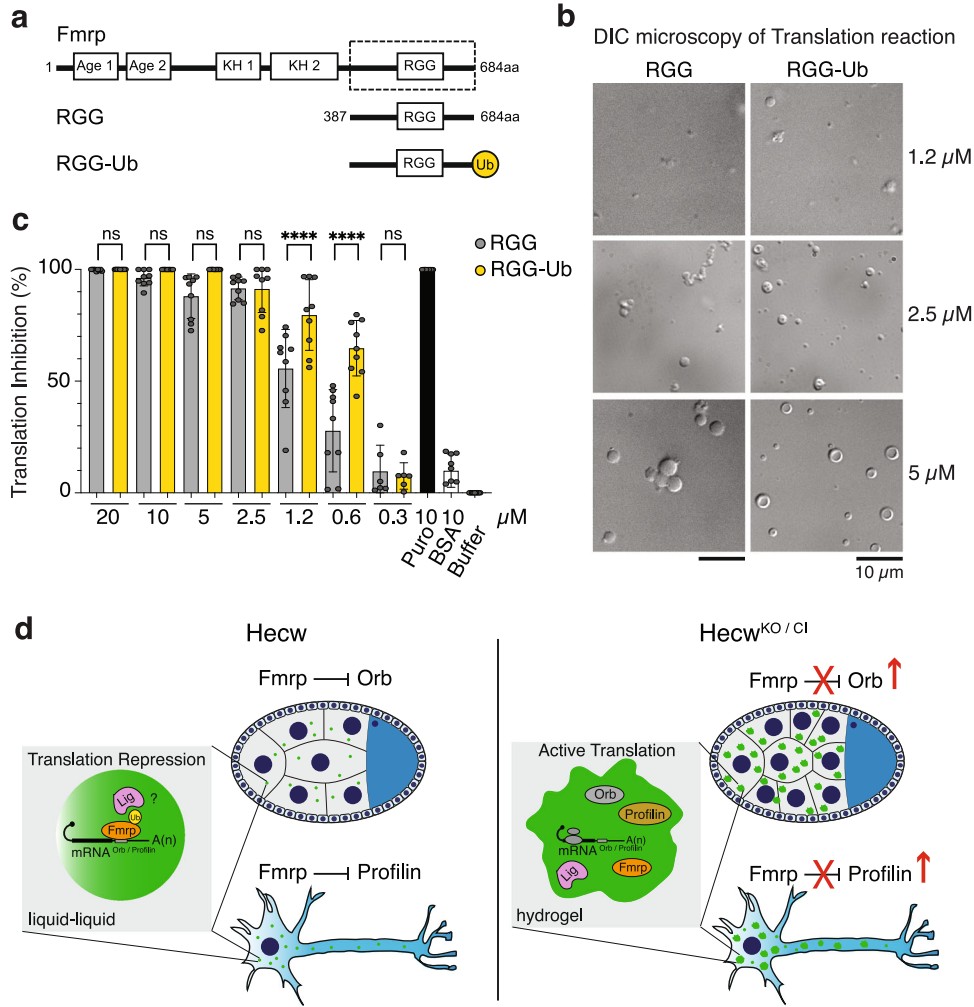

**Fig. 9 Ubiquitin increases Fmrp phase-separation propensity and translational repression. a** Schematic representation of the generated constructs: Fmrp[837-684] (RGG) and Fmrp[837-684]-Ub fusion (RGG-Ub). **b** DIC microscopy images of droplets formed at the indicated protein concentration in the translation reaction mixture. Representative images of three independent experimental replicates. Scale bar: 10 μm. **c** Titration of RGG or RGG-Ub into the luciferase translation reaction inhibits translation. Translation of luciferase mRNA was measured by percentage decrease in luminescence in the presence of RGG (light grey), RGG-Ub (yellow), Puromycin (black), and BSA (white) with respect to Buffer. Results are plotted as mean ± SD. ****$P < 0.0001$ by Ordinary One-Way Anova of three independent experimental replicates. Dots represent individual data points. **d** Proposed model of RNP regulation by Hecw. In normal conditions, both in ovary and in neurons, Hecw ubiquitinates Fmrp and contributes to the maintenance of its translational repression activity, possibly through the interaction with the Ub receptor Lingerer (indicated with a question mark in the figure). In the absence of Hecw, RNPs transition to a less dynamic gel-like state, Fmrp translational repression is lost and *Orb* mRNA and *Profilin* mRNA are actively translated.

released by deubiquitination in the oocyte when translation is due to progress. Similar cycles of ubiquitination/deubiquitination have been identified and characterised in other cellular processes[55–57]. Thus, the existence of Ub receptors, effector components capable of recognising Ub, is predicted[11,58]. Supporting this notion, one of the Hecw interactors is the P-body component Lingerer (Supplementary Table 2), which is known to associate with Fmrp and other RNA-binding proteins to modulate their repressive functions[59] and is endowed with a Ub associated (UBA) domain[11] (Fig. 9d). Similarly, the Ub receptor Rngo may also participate in this regulatory mechanism, as loss of Rngo function in the female germline results in defects similar to the ones observed in *Hecw* mutant flies[60].

While our data are fully compatible with this model, it is also conceivable that Ub may impinge on the physical properties of Fmrp inside RNP granules, as we demonstrated in vitro (Fig. 9a–c). Recent studies showed that phosphorylation, methylation, and sumoylation could control Fmrp phase-separation propensity, and regulation of its intrinsically disordered regions[40,61–63]. In the less

mobile *Hecw* mutant RNPs, Fmrp may be disordered or otherwise uncapable of interacting with repressive complexes and ribosomes. Further studies are needed to test the behaviour of ubiquitinated vs non-ubiquitinated Fmrp and define functional links with different PTMs.

**Hecw and disease.** *Hecw* mutant flies show premature aging, neuronal loss, and neuromotor defects, all reminiscent of neurodegenerative processes in humans. Remarkably, evidence links the human ortholog HECW1 to neurodegeneration. Transgenic mice overexpressing HECW1 display neuronal loss in the spinal cord, muscular atrophy and microglia activation[64], and the HECW1 protein has been shown to ubiquitinate mutant superoxide dismutase 1 (SOD1), typical of familial ALS patients[20]. Similar to Hecw, the expression of HECW1 seems to decline with aging[65], suggesting the existence of a positive and protective role for this E3 ligase in the maintenance of neuronal homeostasis.

Our molecular dissection in *Drosophila* supports a mechanism by which Hecw regulates the spatiotemporal translation of Fmrp

targets. Among them, of particular interest is *Profilin*, a known ALS causative gene[39]. Interesting, profilin mutant mouse model replicates the key features of ALS including adult-onset, progressive motor neuron loss accompanied with progressive motor weakness ending in paralysis and death[66]. While the exact role of Profilin mutants at the onset of the disease remains to be established[67], the identification of a specific Ub ligase acting upstream Profilin expression reveals a new avenue to therapeutically investigate this devastating neurological disease.

Beside ALS, it is worth mentioning that FMR1 alterations are at the basis of the Fragile X syndrome, characterised by intellectual disability, autism as well as premature ovarian failure[68]. Based on our findings, we hypothesise that the severity of these disorders may depend on Hecw status both in neurons and in ovaries. Future studies on Hecw/HECW1 in flies and mammalian models are predicted to illuminate new pathogenetic aspects of these diseases.

## Methods

***Drosophila* strains and cell lines**. Flies were maintained on standard flyfood containing cornmeal, molasses and yeast. All experiments were performed at 25 °C, unless differently specified. The following fly strains were used in this study: $y^1w^1$ (Bloomington *Drosophila* Stock Center [BDSC] #1495), nanosGAL4-VP16 (kindly provided by A. Ephrussi), nanosGAL4-VP16 (BDSC #32180), elav-GAL4;UAS-Dicer2 (BDSC #25750), traffic jam-GAL4 (Kyoto stock center, [DGRC]#104055), Me31B::GFP gift of Tim Weil[69], $Fmr1^{\Delta 113M}$ ([BDSC] # 6929), $Orb^{343}$ (BDSC #58477), UAS-Hecw$^{RNAi}$ CG42797 RNAi (Vienna *Drosophila* Stock Centre [VDRC] #104394), UAS-Fmr1 RNAi (Vienna *Drosophila* Stock Centre [VDRC] #8933). The rescue line Hecw$^{KO;;DC504}$ was generated by crossing Hecw mutant flies with the Dp(1;3)DC504 (BDSC #32313) flies, which contain a the CG42797 locus region inserted on the third chromosome. Transgenic flies were generated by injecting the UASstattB-RFP-CG42797 construct into embryos carrying the attP-zh86Fb φC31 docking site (BestGene.inc). Transgenic offspring was screened by eye-colour (white marker) and sequenced. Strain details are reported in Supplementary Table 3. *Drosophila* S2 cells obtained from Invitrogen, were cultured in Schneider's medium (GibCO) supplemented with 1% Glutamine (Euroclone) and 10% of Fetal Bovine Serum (FBS) (Euroclone) and maintained at standard culture conditions (28 °C).

Ethics approval was not required for experiments on invertebrates.

**Hecw$^{CI}$ and Hecw$^{KO}$ and generation by CRISPR/Cas9 editing**. Guide sgRNAs for CRIPR/Cas9 mutagenesis were designed using the MIT CRISPR design tool (http://crispr.mit.edu, Zhang Lab, MIT). Target sequences were cloned in the pBFvU6.2 vector (NIG-fly stock center) between two BbsI restriction sites with the oligos Hecw$^{CI}$ and Hecw$^{KO}$.

The sgRNA constructs were injected into y,w P{nos-phiC31}; attp2 embryos. Transformants sgRNA flies were crossed with $y^2$ cho$^2$ v$^1$; P{nos-Cas9, y+, v+} 3A/TM6C, Sb Tb (DGRC # CAS-0003) to obtain founder animals with both transgenes. Founder males were crossed with compound-X chromosome (BDSC #64) and the potentially mutated chromosomes were recovered from founder animals over FM7. Cas9 and sgRNA elements were removed from the background thanks to selection of v + eye colour. Mutated chromosomes were identified using T7EI assay. Mutations were further characterised by PCR amplification and sequencing of the target region with the primers Hecw$^{CI}$_F and Hecw$^{CI}$_R or Hecw$^{KO}$_F and Hecw$^{KO}$_R.

For alleles details see Supplementary Fig. 2.

**Constructs**. To generate the pUASstattB-RFP-Hecw construct, the Hecw gene was amplified from LD10978 vector (*Drosophila* Genetic Resource Consortium [DGRC]), with the primers CG42797 BglII_F and CG42797 XhoI_R and cloned by enzymatic digestion into a pUASstattB/RFP vector, previously created using the Infusion HD cloning system (Takara Clontech).

To generate the pGEX6P1-Hecw wild-type construct, the full-length Hecw was amplified from pUASstattB-RFP-Hecw with the primers CG42797 EcoRI_F and CG42797 XhoI_R and cloned by enzymatic digestion into a pGEX6P1(GE Healthcare).

The pGEX6P1-Hecw C1394W construct was generated by site-directed mutagenesis according to the QuikChange Site-Directed Mutagenesis Kit protocol (Agilent) using primers C1394W_F and C1394W_R.

To generate pGEX6P1-WW construct, the region containing the two WW domains of Hecw (636-834aa) was amplified from pUASstattB-RFP-CG42797 construct with the primers CG42797 EcoRI WWI_F and CG42797 XhoI WWII_R and cloned by enzymatic digestion into a pGEX6P1(GE Healthcare).

To generate a pGEX6P1-Hecw (1-130aa) construct for use in antibody production, the N-terminal region of the protein was amplified from pUASattB-RFP-CG42797 with the primers CG42797 EcoRI_F and CG42797 XhoI_Nterm_R and cloned by enzymatic digestion into a pGEX6P1(GE Healthcare).

To generate pET43-His-MBP-Fmrp construct, full-length Fmrp was amplified from cDNA of $y^1w^1$ovaries using the primers Fmrp BamH_F and Fmrp EcoR1_R and cloned by enzymatic digestion into a pET43-His-MBP vector (kind gift of Sebastiano Pasqualato, European Institute of Oncology, Milan).

To generate pGEX6P1-Fmrp$^{387-684}$ construct, the region corresponding to the RGG domain of Fmrp was amplified using full-length Fmrp as template with primers 387_dFMRP_F and Fmrp EcoR1_R and cloned by enzymatic digestion into pGEX6P1 (GE Healthcare).

To generate pGEX6P1-Fmrp$^{387-684}$-Ub (RGG-Ub) fusion protein, Fmrp fragment was amplified from full-length construct using the oligos 387_dFMRP_F and dFMRP_NcoI_R and Ub sequence was amplified from pGEX4T3-Ub WT using the oligos NcoI_Ub_F and Ub_SmaI_R.

The two obtained fragments were cloned by enzymatic digestion into pGEX6P1 vector (GE Healthcare).

Full sequence of the primers used in this study are reported in Supplementary Table 4. All constructs were sequence-verified. The remaining constructs were previously described[19].

**RNA extraction and qPCR**. RNA was extracted from *Drosophila* tissues with Maxwell RSC simplyRNA Tissue kit (Promega) and quantitative PCR (qPCR) analysis was performed with the following TaqMan Gene Expression Assay (Thermo Fisher Scientific): Dm01837441_g1 (CG42797/Hecw), Dm01841193_g1(gurken), Dm02136373_m1(Fmr1), Dm02136342_g1(orb), Dm02134538_g1 (oskar).

**Lifespan assay**. 1 to 3-day-old flies were kept at 25 °C or 29 °C at a density of 25 flies/vial (4 vials/genotype), with mixed-sex groups in standard cornmeal food to minimise the influence of genetic background, environment, nutrition and mating conditions[70,71]. Flies were flipped and scored every two/three days for survivorship. The assay was repeated twice, and data were analysed with PRISM (GraphPad software). Survival fractions were calculated with product limit Kaplan–Meier method and log-rank test was used to evaluate the significance of differences between survivorship curves.

**Climbing assay**. Fifteen flies/genotype were placed in a 9 cm plastic cylinder. After a 30 s rest period, flies were tapped to the bottom of the cylinder. Negative geotaxis was quantitated by counting the number of flies that can cross a 7 cm threshold during a 15 s test period. The climbing index was calculated as the number of succeeding flies over the total. The test was performed on 6–9 independent groups of animals for each genotype, as described in the legends. The climbing ability was measured the first day of life and monitored every 5 days. Significant differences between Hecw mutants and control started to emerge after the 14th day of the lifespan.

**Fertility assay**. Fertility was assessed by counting the eggs laid in 24 h by 20-day-old flies. 6 females and 3 males per genotype were kept in a cage on a 3 cm molasses plate with fresh yeast for a 24 h egg collection after 24 h of adjustment to the cage. The test was performed on three groups/genotype. The hatching rate was calculated as the number of eggs hatched into larvae in 3 days over the total number of eggs.

**Immunostaining and treatments**. Fly ovary dissection and staining was performed as previously described[72]. Briefly, ovaries were fixed in 4% paraformaldehyde (PFA) in PBS (HIMEDIA) for 20 min at room temperature, permeabilised in 1% Triton X-100 in PBS for 20 min, followed by 1 h block in PBST (PBS-Triton X 0.1%) containing 5% (w/v) bovine serum albumin (BSA). Primary antibodies were incubated overnight at 4 °C, secondary antibodies were incubated for 2 h and DAPI (Sigma) was incubated for 15 min at room temperature. Adult brains (Figs. 1e and 8b-c) were dissected in *Drosophila* S2 medium with 10% of FBS (Euroclone) and processed as described before. Ovary microtubule detection (Supplementary Fig. 5e) was adapted from[73]. Briefly, ovaries were incubated in BRB80-T buffer (80 mmol PIPES pH 6.8, 1 mmol MgCl₂, 1 mmol EGTA, 1% Triton X100) for 1 h at 25 °C without agitation. Then, ovaries were fixed in MeOH at −20 °C for 15 min and rehydrated for 15 h at 4 °C in PBST and blocked for 1 h in PBST containing 2% (w/v) BSA before incubation with primary and secondary antibodies overnight in PBST-2% BSA.

In case of drug treatments, fly ovaries were dissected in *Drosophila* S2 medium with 10% of FBS (Euroclone) and incubated with 2.5% 1,6-hexanediol[74] on an orbital shaker at room temperature for 5 min (Fig. 4d and Supplementary Fig. 8c) prior to fixation.

All tissues were mounted in 20% glycerol, 50 mM Tris, pH 8.4 to avoid mechanical sample deformation. Images were acquired using 40× and 60× objectives on a Leica TCS Sp2 upright confocal microscope. The outlines of the oocytes and the anteroposterior (AP) axis were manually specified.

For S2 immunostaining (Supplementary Fig. 1e), cells were plated on coverslips coated with poly-ornithine. S2 cells were rinsed twice with PBS and fixed in 4% PFA for 15 min, permeabilised with PBST for 20 min. After 30 min incubation in Blocking solution (PBST-1% BSA), coverslips were incubated with primary antibody diluted in PBST 0,1% BSA for two hours at room temperature. After three washes in PBS, cells were incubated with secondary antibodies for 2 h at room temperature. DAPI (Sigma) was incubated for 10 min at room temperature then washed once in PBS. Coverslips were mounted on slides using Mowiol Mounting Medium (70% glycerol in PBS, 1.5% DABCO (1,4-Diazabicyclo[2.2.2]octane from Sigma).

TUNEL staining was performed as previously described[75]. Briefly, fixed brains from 30-day-old flies were permeabilized in 100 mM sodium citrate, 0.3% Triton X-100 PBS at 65 °C for 45 min. Brains were then incubated with TUNEL reagent (In Situ Cell Death Detection Kit, TMR red, Sigma) for 14–16 h at 37 °C in dark humid chamber, washed in PBST and incubated with HOECHST 33342 (Life Technologies, 2 µg/ml in PBS) 10 min at 25 °C. Finally, brains were rinsed in water and mounted on glass slides with Prolog Gold fluorescence anti-fading reagent (Invitrogen). Images were acquired with a Nikon ECLIPSE C1si confocal microscope. To quantify TUNEL$^+$ cells, 200–500 cells/individual from $n = 5$ individuals/genotype were scored using the FIJI Software (https://imagej.net/Fiji version 2.1.0/1.53c).

**Immunohistochemical analysis**. Adult heads of 1-day, 30-day and 60-day-old flies were dissected in PBS and fixed in 4% PFA (HIMEDIA) overnight, at 4 °C. Samples were embedded in 1,2% low-melting agarose: while the agarose solidified, heads were properly oriented. Heads-containing agarose blocks were dehydrated in serial dilutions of ethanol (from 70 to 100%) prior to paraffin embedding with a Leica ASP300 Enclosed Tissue Processor. The paraffin blocks were cut with a Leica RM2125 RTS Microtome into 5 µm frontal sections, stained with haematoxylin–eosin (HE) according to standard procedures, and examined by bright-field microscopy. For each time point, at least five brains/genotype were analysed, and vacuoles with diameter >2 µm were counted over 12/15 brain slices.

**Live imaging of GFP labelled protein**. For ex vivo live recording, ovaries of 3-day old *Me31B::GFP* and *Hecw$^{KO}$;Me31B::GFP* females were dissected and mounted on glass bottom MatTek (35 mm) in Halocarbon oil 27 (Sigma) according to[76]. Live cell imaging experiments were performed on the UltraVIEW VoX spinning-disk confocal system (Perkin Elmer), equipped with an EclipseTi inverted microscope (Nikon), a Hamamatsu CCD camera (C9100-50) and driven by Volocity software (Improvision; Perkin Elmer), using a 40× oil-immersion objective (Nikon Plan Fluor, NA 1.3) and a 488 nm laser. One frame every 10 s was acquired for a maximum of 30 min to avoid phenotypic changes due to stress.

**RNP velocity, size and shape measurements**. Me31B::GFP RNP velocity and size analysis was performed using Fiji (https://imagej.net/Fiji version 2.1.0/1.53c)[77,78]. Particles were tracked with the Manual tracking plugin of Image J and analysed with the Chemotaxis plugin (https://ibidi.com/chemotaxis-analysis/171-chemotaxis-and-migration-tool.html). The cursor was placed at the leading edge of each particle in the direction of the movement and tracked until the particle moved out of the plane of focus or ceased moving. The X and Y position of each particle was recorded. Particle size was measured using the Analyse particle tool, setting the same threshold for all images.

The circularity of the RNPs was quantified by evaluating for each granule the so-called shape factor, which is defined as $4\pi A/P^2$, where $A$ is the projected area of the granule and $P$ its perimeter. The definition is such that the shape factor takes the value 1 for a perfectly round object, while it assumes smaller and smaller values as the object become more elongated or irregularly shaped. Area and perimeter of each RNP in a given ovary were obtained by processing the corresponding confocal image *via* a custom MATLAB® code using the function bwboundaries (with 4 pixels connectivity), applied to a binary image obtained from the original image by thresholding. In the case of time-lapse acquisitions, the same threshold was used for all the images in the sequence.

**Fluorescent recovery after photobleaching (FRAP)**. For FRAP recordings, 3-day old *Me31B::GFP* and *Hecw$^{KO}$;Me31B::GFP* females were prepared as described above. Time series were recorded using a Confocal Spinning Disk microscope (Olympus) equipped with IX83 inverted microscope provided with an IXON 897 Ultra camera (Andor) and a IX3 FRAP module equipped with a 405 nm laser using 60× objective. The system is driven by the Olympus CellSens Dimension 1.18 software (Build 16686). The recording frame rate of the GFP signal, excited by 488 nm laser, was 66 ms. Photobleaching was performed via the 405 nm laser for two cycles of 500 ms on a square region of the cytoplasm (3.04 × 3.04 µm) to bleach GFP signal.

Before bleaching, 15 frames were acquired, obtaining a pre-bleaching reference image $I_0(r)$. After bleaching, time-lapse observation was continued for 1500 frames. We consider the azimuthally-averaged intensity profile of the bleached region $I(r,t) = \langle I_0(r) - I(r,t)\rangle_{|r-r_0|=r}$, where $I(r,t)$ is the image intensity at time t after bleaching, $r_0$ is the center of the bleached region and the symbol $\langle \bullet \rangle_{|r-r_0|=r}$

indicates an average overall pixels in the image at a distance $r$ from the center $r_0$ of the bleached region. The process of fluorescence recovery was monitored by following the progressive widening of the bleached spot over time. For each fixed $t$, $I(r,t)$ was fitted by a Gaussian profile $A(t)\exp\left[-r^2/\sigma(t)^2\right] + B(t)$, where the parameter $\sigma(t)$ provides an estimate of the width of the bleached spot. In all cases, we found that the increase of over time of $\sigma(t)$ was well captured by the simple equation $\sigma^2(t) = \sigma_0^2 + 4Dt$, describing the diffusive-like broadening of a Gaussian concentration profile. From a linear fit of $\sigma^2(t)$ over a time interval of 30 seconds, an estimate of the effective diffusion coefficient $D$ was obtained, as shown in the inset of Supplementary Fig. 5d. Compared to standard data treatment, where the FRAP process is monitored by measuring the average intensity within the bleached region, the described procedure, which is based on the time-resolved estimation of a parameter obtained from the fit of a spatial feature, is more robust with respect to the large local intensity fluctuations in the images due to the randomly-moving, highly-contrasted RNPs, whose appearance in the vicinity of the bleached ROI can significantly alter the shape of the recovery curve. The distributions of the obtained values of $D$ are shown in Supplementary Fig. 5e, showing no statistically significant difference between the wild-type and *Hecw$^{KO}$* nurse cells ($D_{WT} = (0.24 \pm 0.03)$ µm$^2$s$^{-1}$, $D_{KO} = (0.24 \pm 0.02)$ µm$^2$s$^{-1}$).

**Fluorescent loss in photobleaching (FLIP)**. FLIP experiments were performed on the UltraVIEW VoX spinning-disk confocal system (Perkin Elmer) equipped with an EclipseTi inverted microscope (Nikon), provided with an integrated FRAP PhotoKinesis unit (Perkin Elmer) and a Hamamatsu CCD camera (C9100-50) and driven by Volocity software (Improvision; Perkin Elmer). Photobleaching was achieved in a square region of $4 \times 4$ µm by using the 488 nm laser at the maximum output to bleach the GFP signal every 5 s, for a total of 100 bleaching events. Initially, five images were acquired to determine the levels of pre-bleach fluorescence. Images were acquired through a 60× oil-immersion objective (Nikon Plan Apo VC, NA 1.4) every 2.5 s for 8 min.

In order to analyse the fluorescence fluctuations over time in FLIP experiments, two homemade Fiji plugins were developed; one to analyse fluorescence either in selected and manual tracked RNPs or in the cytoplasm and the other one to evaluate the fluorescence in the bleached area and in its neighbourhood regions. During each acquisition, a background intensity $I_B(t)$ and a reference intensity $I_R(t)$ were measured. The amplitude $I_B(t)$ of the background noise was obtained as the average intensity within a small ROI in a corner of the image outside the egg chamber. $I_R(t)$ monitors potential systematic changes in the fluorescent emission not directly related with the FLIP experiment and was calculated as the average intensity within a small ROI inside the sample, far from the bleached region and in a different nurse cell. In all of the experiments, we found that both $I_B(t)$ and $I_R(t)$ were fairly constant over time, with no increasing or decreasing net trends. $I_B(t)$ showed very small fluctuations (with RMS amplitude below 5% of the mean value), while in some cases, the profile of $I_R(t)$ was perturbed by the presence of granules moving in and out the ROI. Thus, we adopted a constant value $I_B$ for the background intensity (obtained as a time average of $I_B(t)$) and we did not apply further corrections for systematic intensity drifts.

In each experiment, we selected N ($6 < N < 8$) ROIs of average area 2.5 µm$^2$ outside of the bleached area (half of them in the cytoplasm, half of them in correspondence to RNPs). The area of a ROI associated with an RNP was kept constant over time, while its position was adjusted to follow the moving granule. For the n-th ROI, the average intensity $I_n(t)$ was measured. The relative intensity was calculated as: $i_n(t) = \frac{I_n(t)-I_B}{I_{n,0}-I_B}$, where $I_{n,0}$ was the average intensity within the ROI before bleaching.

Since the diffused intensity can be larger in the region occupied by the cells compared to the outside of the egg chamber, our estimate of $I_B$ represents a conservative estimate of the background contribution. In general, underestimating the background intensity can lead to a systematic overestimate of all relative intensity. However, we found that: (i) the loss curves in the cytoplasm showed no statistically significant difference (in terms of amplitude and kinetics) between *Hecw$^{KO}$* and wild-type; (ii) the loss curves in the bleaching area showed no statistically significant difference (in terms of amplitude and kinetics) between *Hecw$^K$* and wild-type; (iii) the loss curves of the RNPs in *Hecw$^{KO}$* and wild-type were significantly different, the decay time of wild-type RNPs being about three times faster, independently of particular choice of the background intensity. We explicitly verified this independence by repeating the analysis with different values of $I_B$ (25% smaller than the originally estimated value, 25% larger, randomly chosen in in the interval [original value − 25%, original value − 25%]).

To extract a characteristic decay time $\tau$ from the fluorescence loss curves of the RNPs, we fitted our data to a single stretched-exponential decay $f(t) = e^{-(t/\tau)^\beta}$, with $\beta = 0.75$.

We observed that a single laser exposure was not enough to completely "switch-off" the fluorescence signal within the photobleached ROI (Fig. 6a). In fact, the relative intensity drop was only about 0.5 for both *Hecw$^{KO}$* and wild-type.

**Differential dynamic microscopy (DDM)**. To quantify the mobility of RNPs, time series confocal images were analysed with DDM that enables the tracking-free characterisation of the dynamics in a variety of nano- and micro-systems where the individual objects cannot be resolved or tracked with sufficient accuracy, a typical

scenario of biological samples[79]. Rather than reconstructing trajectories followed by the individual particles during their motion in direct space, DDM extracts quantitative mobility information from the analysis of the temporal correlations of the spatial Fourier transforms of the direct space images[80]. We used DDM to estimate the average mean square displacement $MSD(\Delta t)$ of the RNPs[34].

For each image sequence, DDM analysis was performed over a square ROI of average size 45 μm within the egg chamber. For each time delay $\Delta t$, multiple of the delay $\Delta t_0 = 10s$ between consecutive frames, we calculated the image structure function $D(q, \Delta t) = \left\langle \left| \hat{I}(\boldsymbol{q}, t + \Delta t) - \hat{I}(\boldsymbol{q}, t) \right|^2 \right\rangle$, where $\hat{I}(\boldsymbol{q}, t)$ is the 2D Fourier transform of the image at time $t$, $\boldsymbol{q}$ is the wave vector, $q = |\boldsymbol{q}|$, and the symbol $\langle \bullet \rangle$ indicates a combined temporal and azimuthal (i.e. performed over the orientation of $\boldsymbol{q}$) average. The obtained image structure function can be written as

$$D(q, \Delta t) = A(q)\left[1 - f(q, \Delta t)\right] + B, \quad (1)$$

where $A(q)$ is an amplitude term determined by the optical properties of the sample and by the microscope collection optics, $B$ is a term accounting for the delta-correlated noise in the detection chain, and $f(q, \Delta t)$ is the intermediate scattering function (ISF) [ref. [80]], whose decay with $\Delta t$ mirrors the motion of the moving entities in the image.

As first step of our DDM analysis, we estimated the noise term $B$ from an exponential fit of the tail of $D(q, \Delta t_0)$ for $q > 12$ μm$^{-1}$, a procedure that exploits the fact that $A(q)$ vanishes for large enough $q$ due to the finite numerical aperture of the microscope objective. Once $B$ was known, we proceeded with the determination of the amplitude $A(q)$ as the large $\Delta t$ limit of $D(q, \Delta t)$, a procedure that exploits the fact that $f(q, \Delta t)$ decays almost completely to zero for large $\Delta t$. In our experiments, for which this typically happens for $q > 0.4$ μm$^{-1}$, we estimated $A(q)$ from a stretched-exponential fit of $D(q, \Delta t)$ for $\Delta t > 100$ s (Supplementary Fig. 5h).

Once $A$ and $B$ were known, Eq. (1) was inverted to obtain the ISF $f(q, \Delta t)$, as shown in Fig. 5a. For identical particles with Gaussianly distributed displacements, the particle mean squared displacement can be obtained directly from $f(q, \Delta t)$ via a simple algebraical inversion. In less ideal cases, the following relation still holds[34]:

$$MSD_0(\Delta t) = -\lim_{q \to 0} \frac{4}{q^2} \ln f(q, \Delta t), \quad (2)$$

where $MSD_0(\Delta t)$ is a mean square displacement averaged over the entire population of particles present in the image. In our case, the limit for $q \to 0$ was calculated via a linear extrapolation over the wave vector range $0.4 - 0.8$ μm$^{-1}$, as shown in the inset of Fig. 5b.

The results of this procedure, enabling the automatic determination of the average mean square displacement are shown in Supplementary Fig. 5i for all the egg chambers considered in this study. In all cases, the $MSD(\Delta t)$ displayed a linear dependence on $\Delta t$, indicating a diffusive-like behaviour with negligible persistence, at least at the investigated time scales. The corresponding effective diffusion coefficient $D_0$ was estimated from the linear fit $MSD(\Delta t) = 4D_0\Delta t$. After averaging the obtained values overall examined cells of the same type, we found $D_{0,WT} = 0.023 \pm 0.006$ μm$^2$/s and $D_{0,KO} = 0.021 \pm 0.005$ μm$^2$/s, for wild-type and $Hecw^{KO}$, respectively. These two values are fully compatible within the experimental uncertainty, indicating no significant difference in the RNP dynamics between wild-type and $Hecw^{KO}$ egg chambers.

**Statistics and reproducibility.** Statistical analyses were performed with Prism (GraphPad software version 9.1.2). Unless differently specified, all the statistical significance calculations were determined by using either unpaired Student's $t$ test or the non-parametric Mann–Whitney test, after assessing the normal distribution of the sample with Normal (Gaussian) distribution test. Sample sizes were chosen arbitrarily with no inclusion and exclusion criteria. All the immunoblot analysis were performed at least three times with similar results unless differently specified in the legend. Except for immunofluorescence experiments, the investigators were not blind to the group allocation during the experiments and data analyses.

**Antibodies.** Anti-Hecw was produced in rabbit by immunisation with the N-terminal fragment of Hecw (aa 1-130) protein fused to GST protein (Euro-gentech S.A.). The resulting polyclonal antibody was affinity purified and validated (Supplementary Fig. 1e) on S2 cells depleted of Hecw according to the protocol described in [ref. [81]]. Briefly, for double-stranded RNA production, the following T7- and T3-tagged primers were used to amplify the *Hecw* region of interest:

Hecw$^{KD}$ F: 5′-taatacgactcactataggagaGGATAATTGCCACGATTGGT-3′,
Hecw$^{KD}$ R: 5′-aattaaccctcactaaagggagaGGCGCCAATCGTTTGTG-3′. PCR products were transcribed in vitro with T3 and T7 polymerases (Promega) according to the manufacturer's instructions. To generate double-stranded RNAs (dsRNA), T3 and T7 transcripts were annealed at 68 °C for 15 min and at 37 °C for 30 min. For dsRNA treatments, we starved S2 cells for 30 min in serum-free medium. We then added medium with 20% serum and dsRNA at the concentration of 15 μg/10$^6$ cells and incubated for 96 h on coverslips for IF analysis.

All the other antibodies used in this study are listed here: rat anti-elav antibody (IF 1:50; DSHB 7E8A10), rabbit anti-Hecw (IB 1:250, IF 1:200, IP 4 μg/mg; generated in house), rabbit anti-GST (IP 4 μg/mg; generated in house), Llama GFP-TRAP_A (Chromotek gta-20), rabbit anti-GFP (IB 1:5000; Sigma G1544), mouse anti-Fmrp (IB 1:300, IF 1:300, IP 3 μg/mg; DSHB 5A11-s), mouse anti-Profilin (IB 1:200, DSHB chi-1J); mouse anti-Grk (IB 1:400, IF 1:400; DSHB D12-s), rabbit anti-K63 Ub (IF 1:250; Millipore clone Apu-3 05-1308), rabbit anti-K48 Ub (IF 1:250; Millipore clone Apu-2 05-1307), mouse anti-Orb (IB 1:250, IF 1:250; DSHB 4H8-s), mouse anti-Orb (IB 1:250, IF 1:250, IP 1 μg/mg; DSHB 6H4-s), rabbit anti-Osk (IF 1:2000; Ephrussi lab), mouse anti-Ub (IB 1:300, IP 20 μg/mg; Enzo Life Science clone FK2 BML-PW8810), mouse anti-Ub (IB 1:5; generated in house, clone ZTA10 (He at al., 2016)), rat anti-alpha tubulin (IF 1:100; Bio-Rad MCA78G), mouse anti-tubulin (IB 1:1000; Sigma T5168), TRIC-conjugated phalloidin (IF 1:50; Sigma P1958), HRP-conjugated goat anti-mouse IgG (IB 1:10000; Bio-Rad 1721011), HRP-conjugated goat anti-rabbit IgG (IB 1:10000; Bio-Rad 1706515), Alexa647-conjugated donkey anti-rabbit (IF 1:400; Thermo Fisher A31571), Alexa488-conjugated donkey anti-rabbit (IF 1:400; Thermo Fisher A21206), Alexa647-conjugated goat anti-rat (IF 1:400; Thermo Fisher A21247), Alexa488-conjugated donkey anti-mouse (IF 1:400; Thermo Fisher A21202), Cy3-conjugated donkey anti-rabbit (IF 1:400; Jackson Lab 711-165-152), Cy3-conjugated donkey anti-mouse (IF 1:400; Jackson Lab 715-165-150).

**Protein expression and purification.** GST fusion proteins (GST-Hecw, GST-RGG and GST-RGG-Ub) or HIS-MBP-fusion proteins (HIS-MBP-Fmrp full-length) were expressed in Rosetta cells (Novagen) at 18 °C for 16 h after induction with 500 μM IPTG at an OD$_{600}$ of 0.6. Cell pellets were resuspended in lysis buffer (50 mM Na-HEPES, pH 7.5, 200 mM NaCl, 1 mM EDTA, 0.1% NP40, 5% glycerol, and Protease Inhibitor Cocktail set III (Calbiochem)). Sonicated lysates were cleared by centrifugation at 48,000 × g for 45 min. Supernatants were incubated with 1 ml of Glutathione Sepharose™ 4B beads (GE Healthcare) or with Amylose resin (New England BioLabs) per liter of bacterial culture. After 4 h at 4 °C, beads were washed with PBS and equilibrated in maintenance buffer (50 mM Tris-HCl, pH 7.4, 100 mM NaCl, 1 mM EDTA, 1 mM DTT, 10% glycerol). The His-tagged E1 enzyme Uba1 and His-tagged E2 enzyme Ube2D3 (UBCH5c) used for in vitro ubiquitination were produced as described in [ref. [82]].

GST-Hecw was eluted from Glutathione Sepharose in elution buffer (100 mM Tris-HCl, pH 8, 200 mM NaCl, 1 mM EDTA, 50 mM Glutathione (Sigma)). Fractions containing the desired protein were collected and dialysed against 50 mM Tris-HCl pH 8, 200 mM NaCl, 1 mM EDTA, 10% glycerol, 1 mM DTT and concentrated.

GST-tagged RGG and GST-RGG-Ub were cleave off from the GST tag by addition of 3 C protease overnight at 4 °C in maintenance buffer, directly on beads. Uniform completion of the 3 C protease cleavage reaction was confirmed by SDS-PAGE. After cleavage the beads were extensively washed with maintenance buffer and all the fractions containing cleaved proteins were collected. Combined fractions were concentrated and loaded on Superdex 75 size exclusion chromatography column (GE Healthcare) pre-equilibrated with 25 mM NaPO$_4$ pH 7.4, 50 mM NaCl, 2 mM DTT. Fractions containing the proteins of interest were collected and loaded on HiTrapQ column (GE Healthcare) in the same purification buffer. Flow-through and wash fractions that contain the proteins of interest were combined and concentrated. Eventual RNA contamination is retained by the HiTrap Q column and eluted at high salt concentration. Protein purity was analysed by SDS-PAGE, while protein concentrations were determined by absorbance at 280 nm. The UV absorbance ratio of 260/280 nm < 0.6 was used to verify absence of RNA contamination.

Untagged wild-type Ub (Sigma) was resuspended in maintenance buffer (50 mM Tris-HCl, pH 7.4, 100 mM NaCl, 1 mM EDTA, 1 mM DTT, 10% glycerol) and was purified using a Superdex 75 size exclusion chromatography column (GE Healthcare). Ub mutants K63R and K48R are from Boston Biochem.

**In vitro ubiquitination.** Reaction mixtures contained purified enzymes (20 nM E1, 250 nM purified Ube2D3, 250 nM GST-Hecw), 300 nM substrate (Amylose resin bound His-MBP-Fmrp) and 1.25 μM Ub either wild-type or mutants in ubiquiti-nation buffer (25 mM Tris-HCl, pH 7.6, 5 mM MgCl$_2$, 100 mM NaCl, 0.2 μM DTT, 2 mM ATP) were incubated at 30 °C for 60 min. Samples were then centrifuged in order to separate the pellet (containing the ubiquitinated substrate) from the supernatant containing the enzymes and the free Ub chains eventually produced. Pellets were washed four times in WASH buffer (50 mM Tris-HCl pH 7.4, 300 mM NaCl, 0.1% Triton-X100, 5% glycerol, 1 M UREA) before loading on SDS-PAGE gel. Detections were performed by immunoblotting using the anti-Ub antibody according to the protocol described in [Ref. [82]]. Membranes were stained with Coomassie after immunoblotting to show equal loading.

**In vitro translation assay.** In vitro translation assay was performed as described in [ref. [40]]. Briefly, translation of luciferase mRNA (Promega) was performed with nuclease-treated rabbit reticulocyte lysate (Promega). Each reaction mix (30 μl) contained the following components: 12.6 μl of rabbit reticulocyte lysate, 0.5 μl of Luciferase mRNA (1 mg/ml), 0.3 μl of amino acid mixture minus leucine (1 mM), 0.3 μl of amino acid mixture minus methionine (1 mM), and 16.3 μl of proteins,

puromycin (Sigma) or buffer (25 mM NaPO4 pH 7.4, 50 mM NaCl, 2 mM DTT). Reactions were assembled at different concentration of RGG or RGG-Ub (20-10-5-2.5-1.2-0.6-0.3 µM) or BSA (10 µM). All in vitro translation reactions were prepared on ice, followed by incubation at room temperature for 3 h before luminescence measurement. Luciferase translation was measured with luciferase assay system (Promega) by mixing 75 µl of luciferase substrate with 3 µl of unpurified translation mixture in a black 96-well plate (Corning). End-point luminescence measurements were carried out in triplicate using Victor3 multilabel plate reader (Perkin Elmer).

**DIC imaging of phase separation samples**. 10 µl of translation mixtures of the in vitro reactions described above were transferred on imaging chambers prepared as described in [ref. [83]] and visualised for the appearance of liquid droplets with DIC microscopy. Images were acquired with DeltaVision Elite microscope (GE Healthcare) equipped with a Cool SNAP HQ2 camera using UPlanSApo 100× 1.4NA oil-immersion objective and driven by Softworx 6.5.2 version. Phase separation in different conditions was defined by the observation of round droplets. All images represent a single focal plane focused onto the surface of the glass slide. Images were processed with FIJI Software (https://imagej.net/Fiji version 2.1.0/1.53c).

**Western blots and immunoprecipitations**. *Drosophila* tissues, collected as described in [ref. [72]], were homogenised with a pestle and incubated for 20 min on ice in RIPA buffer (50 mM Tris-HCl pH8, 150 mM NaCl, 1% NP-40, 1% sodium deoxycholate, and 0.1% SDS) supplemented with a protease inhibitor cocktail (CALBIOCHEM), clarified by centrifugation and analysed by immunoblotting.

For co-immunoprecipitation experiments (Fig. 7a and Supplementary Fig. 7a), *Drosophila* ovaries were lysed in JS buffer (50 mM Tris-HCl pH 7.6, 150 mM NaCl, 10% glycerol, 1% Triton X-100, 1.5 mM MgCl₂, 5 mM EGTA,) supplemented with protease inhibitors (Calbiochem). After extensive washes with JS buffer, beads were re-suspended in Laemmli-buffer and proteins analysed by SDS-PAGE and immunoblotting.

In case of drug treatments, fly ovaries were dissected in *Drosophila* S2 medium with 10% of FBS (Euroclone) and incubated with 50 µM MG132 on an orbital shaker for 2 h (Supplementary Fig. 7e) prior lysis.

For immunoprecipitation of ubiquitinated proteins, 40 µl of Agarose-TUBEs 2 (UM402, LifeSensors) previously equilibrated in JS buffer, were incubated overnight with 1 mg of ovary lysate on an orbital shaker at 4 °C. After overnight incubation, the lysate was spin down and the supernatant was re-incubated with additional 30 µl of TUBEs for 1.5 h at 4 °C. Beads were then washed, re-suspended in Laemmli-buffer, loaded on SDS-PAGE and analysed by immunoblotting.

**LC-MS/MS analysis**. For identification of Hecw interactors, a GST fusion construct encompassing the two WW domains of Hecw (636-834aa) was used. Briefly, 2 µM of GST proteins were incubated with 1 mg of S2 lysate for 2 h at 4 °C in YY buffer (50 mM Na-HEPES pH 7.5, 150 mM NaCl, 1 mM EDTA, 1 mM EGTA, 10% glycerol, 1% triton-100). After four washes of the GST proteins with YY buffer, specifically bound proteins were resolved on 4–12% gel (Invitrogen) and detected by Coomassie staining. Samples were processed for MS analysis according to the STAGE-diging protocol described in[84]. Peptide mixtures were acidified with 100 µL of 0.1% formic acid (FA, Fluka) and eluted by the addition of 100 µL of 80% ACN, 0.1% FA, followed by a second elution with 100% CAN. All of the eluate peptides were dried in a Speed-Vac and resuspended in 15 µL of solvent A (2% ACN, 0.1% formic acid) and 4 µL were injected for each analysis on the Q Exactive HF-X Hybrid Quadrupole-Orbitrap (Thermo Fisher Scientific). Peptides separation was achieved with a linear gradient from 95% solvent A (2% ACN, 0.1% formic acid) to 50% solvent B (80% acetonitrile, 0.1% formic acid) over 33 min and from 50 to 100% solvent B in 2 min at a constant flow rate of 0.25 µL/min, with a single run time of 45 min. MS data were acquired using a data-dependent top 12 method, the survey full scan MS spectra (300–1650 Th) were acquired in the Orbitrap with 60000 resolution, AGC target 3e6, IT 20 ms. For HCD spectra resolution was set to 15000, AGC target 1e5, IT 80 ms; normalised collision energy 28 and isolation width of 1.2 m/z. Raw data were processed using Proteome Discoverer (version 1.4.0.288, Thermo Fischer Scientific). MS² spectra were searched with Mascot engine against an in-house *Drosophila Melanogaster* Db revised version (according to[85]). Scaffold (version Scaffold_4.3.4, Proteome Software Inc., Portland, OR) was used to validate MS/MS based peptide and protein identifications. Peptide identifications were accepted if they could be established at a probability >95.0% by the Peptide Prophet algorithm[86] with Scaffold delta-mass correction. Protein identifications were accepted if they could be established at a probability >9.0% and contained at least two unique high confident peptides.

**Reporting summary**. Further information on research design is available in the Nature Research Reporting Summary linked to this article.

## Data availability
The mass spectrometry raw datasets generated in this study have been deposited in PRIDE database and can be accessed through ProteomeXchange with the following Project Name: Pulldown CG42797 vs S2 cell lysate and Project accession: "PXD025809". Full list of the specific interactors is provided in Supplementary Table 2. Source data are provided within this paper. Source data are provided with this paper.

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

## Acknowledgements

We thank Sebastiano Pasqualato for DNA constructs, Ilaria Busi for initial generation of Hecw$^{CI}$ mutants and Stefano Confalonieri for reviewing the Mascot *Drosophila melanogaster* proteins database. We thank Paolo Soffientini, Federica Pisati, Valentina Dall'Olio and Laura Tizzoni at Cogentech facilities (Milan, Italy) for support in mass spectrometry, immunohistochemistry and qPCR analysis. We thank BDSC, DGRC, and DSHB for providing valuable *Drosophila* reagents to us and the community. We are grateful to Wessen Maruwge for English language editing. This work was supported by the Italian Ministry of Education, Universities and Research (PRIN 20108MXN2J to S.P.); the CARIPLO foundation (2017-0746 to El.M.); the Associazione Italiana per

Ricerca sul Cancro, (Investigator grant 2017-20661 to T. V., MFAG 2018-22083 to F.G.). Valentina Fajner's work is supported by the Associazione Italiana per la Ricerca sul Cancro. V.F. was and S.S. is a PhD student of the European School of Molecular Medicine (SEMM).

## Author contributions
Conceptualisation: V.F., El.M., T.V. and S.P; V.F. performed all the *Drosophila* experiments; El.M. generated all the biochemical data and supervised work by S.S. and V.F; A.O., Em.M., and D.P. aided in all the imaging acquisition and FLIP experiments; F.G. and R.C. analysed and quantitatively elaborated the live images; F.N. performed tunnel experiment; G.C. prepared genetic crosses for revision; El.M. and T.V. contributed to planning and interpretation of data; S.P. coordinated the team, designed and supervised the project and wrote the paper with contributions from all authors.

## Competing interests
The authors declare no competing interests.
