## [Peer Review File · Nature Communications]

REVIEWER COMMENTS

Reviewer #1 (Remarks to the Author):

Ribonucleoprotein (RNP) granules are key membrane-less organelles in polarized cells including germ cells and neurons. Recent studies have demonstrated that the formation of RNPs may be driven by biomolecular liquid-liquid phase separation. In this manuscript, the authors proposed that the E3 ubiquitin ligase Hecw regulates the liquid state and the diffusive exchange in and out of RNPs in *Drosophila* oogenesis and neurons, possibly through its ubiquitination on a key RNP component Fmrp. They found that depletion of Hecw caused altered Orb expression in nurse cells and Profilin in neurons, resulting in gel-like transition of RNPs in germ cells and defective oogenesis, and age-dependent neuronal degeneration. They suggested that Hecw ubiquitinates Fmrp to maintain mRNAs of Orb and Profilin in a repressed state. Overall, this paper presents an interesting finding for Hecw ligase activity in RNP regulation. However, there are a number of concerns that have to be addressed to make a convincing case, which are outlined in the following.

Major comments:

1. The authors discovered that the reduced ubiquitination of Fmrp by Hecw impaired its translational repressor activity over Orb in nurse cells and Profilin in neurons. They also observed hecwKO-induced enlargement and transition to a less fluid, gel-like state of Me31B-positive RNPs. What's the underlying molecular mechanism? The model proposed in Fig7e is just speculation and needs in vitro demonstration. Fmrp has been shown to undergo liquid-liquid phase separation in vitro, which is linked to its translational repressor activity. Hecw directly interacts with Fmrp through its WW domains. Besides, the authors observed that the Orb protein which was undetectable in the nurse cells granules, was clearly present in the larger Hecw-depleted RNPs, suggesting a lack of local translational repression and recruitment of Orb into the RNPs. Thus, can Orb or Profilin be recruited into the Fmrp liquid phase, which further promotes its gel-like transition in vitro? What's the direct physiologic relevance of Hecw mediated K63-Ubiquitination of Fmrp? Does ubiquitination change the physical property of Fmrp condensates? How? In a Lingerer dependent way?
2. It seems to me that the FRAP and FLIP data are contradictory. The FRAP assay recorded the recovery process of photobleached RNPs which directly equilibrated with neighboring regions, and the FLIP assay recorded the decrease of selected neighboring regions which directly equilibrated with the photobleached RNPs. Both assays provided information about the dynamic properties of RNPs and the results should be consistent. However, the author claimed that the FRAP profiles in Me31B::GFP and HecwKO Me31B::GFP fly ovaries were comparable, whereas the RNPs were more dynamically exchanging with surroundings in the wt compared to the HecwKO mutant flies. The authors need to provide a detailed explanation.
3. Some results were not convincing. Supplementary Fig.5c suggested that the fluorescence loss curves in the cytoplasm regions were comparable between control and HecwKO, however, the fluorescence loss curves and images in the cytoplasm regions were quite distinct between control and HecwKO in Supplementary Fig.5a.
4. Supplementary Fig.5b was not done properly. The regions to be bleached (yellow rectangle) should be comparable in the control and HecwKO flies, but not as the current data in which the former had no RNP whereas the latter contained large RNPs. It is predictable that the exchanging rates in two situations were quite different.

Reviewer #2 (Remarks to the Author):

Hecw controls oogenesis and neuronal homeostasis by promoting the liquid state of ribonucleoprotein particles.

In this manuscript, Fajner et. al., identified CG42797/Hecw as a *Drosophila* HECT E3 ubiquitin ligase. By generating Hecw knockout (KO) and catalytically-inactive (CI) mutant flies, the authors demonstrate that Hecw has pivotal roles in oogenesis and on preventing neuronal degeneration especially in aging flies. To understand the mechanism of HECT E3 function, the authors identified dFMRP as a substrate of Hecw and genetically demonstrated that the loss of Hecw compromises dFMRP function in regulating Orb level in germ cells and Profilin in neurons, which may account for neurodegeneration and fertility phenotype observed in Hecw mutants, respectively. Further they show that Hecw could promote the liquid state of RNP granules in germ cells and neurons by interacting with, and potentially ubiquitinating the components within the granules.

Major points:

1. The title and final model (Fig. 7e) of the manuscript are suggesting the liquid state of RNP granule, maintained by Hecw, might affect the translational state of the mRNA inside the granules and thus lead to the oogenesis and neurodegeneration phenotypes. But no data are provided to support this idea. The gelation of RNP granules and deregulation of translation could equally well be coincident but independent consequences of loss of Hecw function based on the data provided.
2. The mechanism of granule liquid state maintenance by Hecw requires more experiments to understand. It is unclear whether the gel-like transition was caused by the loss of the ubiquitination by Hecw or other potential ubiquitination-independent functions of Hecw. To resolve this question, authors should examine the liquid/gel state of granules in Hecw CI mutant in addition to their existing data from KO flies (Fig. 4 and Fig. 5).
3. If the liquid state is indeed maintained by the ubiquitination, is it due to the ubiquitination of structural components like ME31B or indirectly caused by an accumulation of mis-expressed proteins like Orb? The authors could address this by asking whether ME31B is ubiquitinated by Hecw in vivo and whether reducing Orb by RNAi or using a heterozygote can suppress the gel like state.
4. One notable finding of this study is identifying dFMRP as a substrate of Hecw. Consistently, Hecw mutant flies phenocopy dFMRP loss of function mutants in oogenesis. This suggests a novel and intriguing mechanism by which poly-ubiquitination potentiates dFMRP function as a translational repressor. However, proving direct evidence for this conclusions, will require the identification of ubiquitination sites on dFMRP and making the corresponding mutants for functional in vivo study. Alternatively, authors should at least corroborate their Hecw-dFMRP-Orb axis model by testing whether orb heterozygosity (using alleles like orb343 or orbdec) or over-expressing fmr1 can suppress Hecw-KO/CI oogenesis phenotypes.
5. To add more direct support to their conclusion that Hecw regulates Orb/Profilin level specifically through dFMRP, the author need to over-express Hecw in fmr1 homozygous mutant background (using viable alleles like fmr13 or fmr1 Δ 113). Orb level should not be decreased by Hecw over-expression when dFMRP is absent.

Minor points:

1. The genetic nature of the rescuing allele DC504 should be explained in the manuscript or in the Figure 2 legend.
2. Authors noted that both, the defective egg chamber phenotype and the orb mis-expression phenotype have limited penetrance (20%-30% egg chambers). How do these two phenotype correlate? Do defective egg chambers tend to have orb mis-expression or puncta in nurse cells?
3. The in vitro ubiquitination experiment presented in Figure 6d needs to include a control lane without

Fmrp as the poly-ubiquitin chain signal might come from Hecw self-ubiquitination and non-specific binding to NiNTA beads.

4. Fig. 6h and 6i, the authors should note the penetrance of Grk/Osk mis-localization phenotypes.

Reviewer #3 (Remarks to the Author):

review of manuscript NCOMMS-20-31849, Fajner et al.

This paper deals with the functional characterization of the *Drosophila* Hecw ubiquitin ligase during oogenesis and in neurons. The authors generated knockout and catalytically inactive mutant alleles of Hecw and analyzed the phenotypes of the mutants, in particular during oogenesis. The mutants are homozygous viable but show reduced lifespan and impaired climbing behavior, indicative of motoneuron function impairment. A detailed analysis of oogenesis in the mutants revealed that egg chambers displayed a variety of morphological defects, ranging from abnormal number of nurse cells, the presence of two oocytes in one egg chamber, ring canal defects and in general reduced numbers of laid eggs. Intriguingly, the mutants show abnormally sized and shaped RNP granules in germ line cells that contained Orb protein, which is normally confined to the oocyte and is degraded in nurse cells. By use of live imaging analysis and fluorescence bleaching assays the authors could show that the normally liquid-like state of the RNP granules was changed to a more solid, gel-like state in the mutants. Using co-IP and GST binding assays, the authors identified fragile-X mental retardation protein (FMRP), a translational repressor crucial for repression of Orb translation in nurse cells, as a binding partner and ubiquitination target of Hecw. FMRP also has a function in neurons where it regulates the levels of the actin-associated protein profilin. Indeed, profilin levels in Hecw mutant brains were also abnormal, pointing to a general function of Hecw in regulating protein translation via FMRP.

The data in this paper are of excellent quality and the manuscript is well organized and well written throughout. The story told in the manuscript is of general interest and novelty, as RNP granules and the human homolog of FMRP are involved in the pathogenesis of Amyotrophic lateral sclerosis (ALS) a devastating deadly disease. This paper now shows for the first time in the model organism *Drosophila* that a ubiquitin ligase can regulate the biophysical properties and functions of RNP granules and of the FMRP protein, which may open new avenues in the research on ALS. I recommend publication of the manuscript in *Nature Communications*, provided that a few minor points will be addressed.

Suggestions to the authors:

1. Supplementary Table 2: This table just lists the names of several proteins that were identified as binding partners of GST-Hecw in a pull-down assay analyzed by mass spectrometry. Here I suggest to publish the full Excel file with proteins identified by mass spec as supplementary file, including significance values, numbers of identified peptides etc., so that the reader can decide on the significance of the potential interaction.
2. Figure 6: In panels b, h and i immunofluorescence images of oogenesis phenotypes are shown that were observed in FMRP mutants (b) and in Hecw mutants regarding the mislocalization of Gurken (h) and Oskar (i). The frequencies of these phenotypes in comparison to wild type needs to be quantified.
3. Discussion, page 10: The authors discuss that ubiquitin receptors, proteins reading the ubiquitin modifications on proteins such as FMRP, are likely to be involved in the regulation of RNP properties. They mention the ubiquitin receptor Lingerer, which was found in the GST pulldown assay as a potential binding partner of Hecw. They should also mention and discuss the ubiquitin receptor Rings

lost (Rngo), which shows strikingly similar oogenesis phenotypes as Hecw mutants in germ line clones (Morawe et al., 2011, J Cell Biol 193, 71)

Reviewer #1 (Remarks to the Author):

Ribonucleoprotein (RNP) granules are key membrane-less organelles in polarized cells including germ cells and neurons. Recent studies have demonstrated that the formation of RNPs may be driven by biomolecular liquid-liquid phase separation. In this manuscript, the authors proposed that the E3 ubiquitin ligase Hecw regulates the liquid state and the diffusive exchange in and out of RNPs in *Drosophila* oogenesis and neurons, possibly through its ubiquitination on a key RNP component Fmrp. They found that depletion of Hecw caused altered Orb expression in nurse cells and Profilin in neurons, resulting in gel-like transition of RNPs in germ cells and defective oogenesis, and age-dependent neuronal degeneration. They suggested that Hecw ubiquitinates Fmrp to maintain mRNAs of Orb and Profilin in a repressed state. Overall, this paper presents an interesting finding for Hecw ligase activity in RNP regulation. However, there are a number of concerns that have to be addressed to make a convincing case, which are outlined in the following.

Major comments:

1. The authors discovered that the reduced ubiquitination of Fmrp by Hecw impaired its translational repressor activity over Orb in nurse cells and Profilin in neurons. They also observed *hecw*KO-induced enlargement and transition to a less fluid, gel-like state of Me31B-positive RNPs. What's the underlying molecular mechanism? The model proposed in Fig7e is just speculation and needs *in vitro* demonstration. Fmrp has been shown to undergo liquid-liquid phase separation *in vitro*, which is linked to its translational repressor activity. Hecw directly interacts with Fmrp through its WW domains.

R: We thank the Reviewer for pointing out this important aspect that was also raised by Reviewer # 2. To establish a causal role for Fmrp-ubiquitination and its translational repressor activity, we took advantage of an *in vitro* system that was originally set up in the Forman-Kay lab {Tsang, 2019 #57}. With this elegant translation assay coupled to liquid droplets formation, the authors demonstrated that the human Fmrp RGG domain is the minimal region of the protein able to both sustain translation inhibition and undergo LLP transition. We cloned the corresponding *Drosophila* fragment and we also generated a chimeric protein by fusing a Ub moiety at its C-terminus. Notably, the RGG-Ub fusion protein shifted the translation inhibition effect at a lower protein concentration, when compared to RGG alone, and displayed a parallel increase of the phase separation of the RNA-protein complex. These important results are now reported in the new Fig. 8a-c and described in the revised manuscript.

Moreover, we saw a clear recruitment of the bacterially purified RFP-Hecw protein to the surface of the droplets formed by the Fmrp RGG domain that contains the putative WW interaction motif. We intend to follow up on this intriguing result in our future studies, but we add an example below for the Reviewer's eyes.

Besides, the authors observed that the Orb protein which was undetectable in the nurse cells granules, was clearly present in the larger Hecw-depleted RNPs, suggesting a lack of local translational repression and recruitment of Orb into the RNPs. Thus, can Orb or Profilin be recruited into the Fmrp liquid phase, which further promotes its gel-like transition in vitro? What's the direct physiologic relevance of Hecw mediated K63-Ubiquitination of Fmrp? Does ubiquitination change the physical property of Fmrp condensates? How? In a Lingerer dependent way?

R: All of the questions listed by the Reviewer are very interesting and represent *per se* new lines of research. Deciphering how the Ub chain specificity affects the phase behavior of condensates is certainly one of the main questions we would like to address in the future. Besides the *in vitro* characterization previously discussed, we tried to produce and purify the Orb protein. Unfortunately, this turned out to be extremely tricky, as the protein is poorly soluble and aggregates. Therefore, we could not verify the possible recruitment of Orb into the RNPs that we scored *in vivo*. Thus, we turned to a genetic validation step of our findings, which allowed us to conclude that the gel-like transition we observed in the absence of Hecw activity is independent of Orb recruitment to the enlarged RNPs (see our reply to Reviewer # 2 and pictures therein).

The possible role of Lingerer was presented with a question mark in the model of Fig. 7e to graphically guide the readers throughout the discussion and was not meant as a final recap of what we discovered. We are ready to eliminate Lingerer from the new Fig. 8d if required, although we are hesitant as it suggests an interesting and novel scenario in terms of regulation.

2.It seems to me that the FRAP and FLIP data are contradictory. The FRAP assay recorded the recovery process of photobleached RNPs which directly equilibrated with neighboring regions, and the FLIP assay recorded the decrease of selected neighboring regions which directly equilibrated with the photobleached RNPs. Both assays provided information about the dynamic properties of RNPs and the results should be consistent. However, the author claimed that the FRAP profiles in Me31B::GFP and HecwKO Me31B::GFP fly ovaries were comparable, whereas the RNPs were more dynamically exchanging with surroundings in the wt compared to the HecwKO mutant flies. The authors need to provide a detailed explanation.

R: We apologies for not having conveyed a clear message. The results obtained with different experimental approaches (in particular DDM, FRAP and FLIP) are fully compatible with each other and present a rather consistent picture. We acknowledge that the organization of the figures was potentially misleading. We have now revised the manuscript to more accurately describe the assays used and better highlight our findings. Here below we also add a brief explanation that we hope will clarify the reviewer's questions. An important point that was not previously spelled-out clearly is that, in the FRAP experiments the bleached region does not include any RNP (Supplementary Fig. 5i,l). As a matter of fact, the fast trafficking of the RNPs precluded the possibility to probe the RNPs directly by FRAP and this is precisely the reason why we also employed FLIP. Thus, the FRAP experiments were primarily aimed at probing the transport properties of the fluorescent protein Me31B::GFP in the cytoplasm. The fact that we do not find significant differences between CTRL and a Hecw KO egg chamber supports the assumption that cytoplasmic transport is not affected by Hecw depletion. These results are fully consistent with the ones obtained from the FLIP experiments, when the ROIs outside of the bleached area, devoid of any RNP, are considered. Even in this case, the loss curves show no significant difference between CTRL and a Hecw KO (new Fig. 5d and Supplementary Fig. 6d-f). The sole and striking difference we scored in Hecw KO egg chambers is related to the FLIP experiments where the ROIs were centered on RNPs. As

better explained in the next reply, a marked difference is visible in the loss curves of CTRL and Hecw KO egg chambers (Fig. 5c)

Taken together, these results indicate that the lack of Hecw activity severely impacts on the aggregation state of the RNPs, while it has negligible effects on the transport of cytoplasmic Me31B.

3. Some results were not convincing. Supplementary Fig.5c suggested that the fluorescence loss curves in the cytoplasm regions were comparable between control and HecwKO, however, the fluorescence loss curves and images in the cytoplasm regions were quite distinct between control and HecwKO in Supplementary Fig.5a.

R: We assume that the figure the Reviewer is referring to is Fig. 5 of the original manuscript. The curves shown in former Fig. 5a are representative results of specific FLIP experiments (performed on a CTRL and a Hecw KO egg chamber, respectively), where rather large ROIs are considered (colored squares). Each ROI contains many RNPs, and both RNPs and cytoplasm contribute to the fluorescent signal. As the Reviewer correctly points out, we indeed observed a significant difference between CTRL and Hecw KO in the so-obtained fluorescence loss curves. However, this difference cannot be unambiguously attributed to a different rate of exchange in the granules, as it could be also due to altered protein transport in the cytoplasm, which was indeed the reason why we performed the subsequent experiments.

The results showed in current Fig. 5 are all obtained from a more refined analysis of the FLIP experiments that allowed us to address this point. In this case, we consider smaller ROIs, each containing either a single RNP or a granule-free portion of the cytoplasm of a comparable size. All of the selected ROIs are outside of the bleached area, at approximately the same distance from its center. The results showed a marked difference in the RNPs loss curves obtained for CTRL and Hecw KO, clearly indicating that protein exchange is significantly impaired in Hecw KO cells (Fig. 5c). On the contrary, no significant difference is found in the loss curves obtained from cytoplasmic ROIs, indicating that the protein diffusion in the cytoplasm is not altered in Hecw KO cells (Fig. 5d). Thus, these results appear to be fully consistent with the ones shown in former Fig. 5a.

To make the presentation of these results clearer and more effective, we have taken the following actions:

- We have extensively revised the manuscript describing these experiments.
- We moved former panel 5a to the new Supplementary Fig. 6 (panel a).
- We added representative ROIs of RNPs-free regions in the images shown in panel 5a.
- We modified the legend and added explanatory titles and plots tags in panel 5b, 5c and 5d, to show more clearly which part of the egg chamber is considered in each case.
- We added a few representative movies of FLIP experiments (new Supplementary movies 4-7).

4. Supplementary Fig.5b was not done properly. The regions to be bleached (yellow rectangle) should be comparable in the control and HecwKO flies, but not as the current data in which the former had no RNP whereas the latter contained large RNPs. It is predictable that the exchanging rates in two situations were quite different.

R: We apologize for the lack of clarity. The Reviewer is right in pointing out that, at least as far as the fluorescent signal *within* the bleached area is considered, regions containing or not containing RNPs can behave differently. However, we are measuring the

fluorescence intensity in RNP granules (current Fig. 5c) and granule-free portion of the cytoplasm (current Fig. 5d), *outside* and far from the bleached area where we do not expect any systematic effect on the loss curves. To avoid any selection bias, the area to be bleached is randomly selected within the egg chamber. Thus, the probability that one or more granules are present within the bleached area is the same for CTRL and Hecw KO. We have clarified the procedure in the methods section. We acknowledge that the choice of the representative images in former Fig. 5b could be misleading, as it may suggest that a systematic bias was present in the selection of the ROI to be bleached. For this reason, we have changed the representative image of CTRL and added a few quantified movies as supplementary data (Supplementary movies 4-7).

Reviewer #2 (Remarks to the Author):

In this manuscript, Fajner et. al., identified CG42797/Hecw as a Drosophila HECT E3 ubiquitin ligase. By generating Hecw knockout (KO) and catalytically-inactive (CI) mutant flies, the authors demonstrate that Hecw has pivotal roles in oogenesis and on preventing neuronal degeneration especially in aging flies. To understand the mechanism of HECT E3 function, the authors identified dFMRP as a substrate of Hecw and genetically demonstrated that the loss of Hecw compromises dFMRP function in regulating Orb level in germ cells and Profilin in neurons, which may account for neurodegeneration and fertility phenotype observed in Hecw mutants, respectively. Further they show that Hecw could promote the liquid state of RNP granules in germ cells and neurons by interacting with, and potentially ubiquitinating the components within the granules.

R: We are very grateful to Reviewer # 2 for the critical reading of our manuscript and the numerous useful suggestions that we have now implemented to improve our study.

Major points:

1. The title and final model (Fig. 7e) of the manuscript are suggesting the liquid state of RNP granule, maintained by Hecw, might affect the translational state of the mRNA inside the granules and thus lead to the oogenesis and neurodegeneration phenotypes. But no data are provided to support this idea. The gelation of RNP granules and deregulation of translation could equally well be coincident but independent consequences of loss of Hecw function based on the data provided.

R: We thank the Reviewer for drawing our attention to this important point. In the revised manuscript, we provide new mechanistic insights that sustain the causal link between the two phenotypes depicted in the final model (now in Fig. 8d).

As mentioned before in the response to Reviewer # 1, we optimized an *in vitro* reconstituted model of translational assay and biomolecular condensate based on the intrinsically disordered C-terminal region of FMRP (RGG domain) originally set up by the group of Forman-Kay. With that method we were able to prove that the addition of Ub molecule to the FMRP minimal region is sufficient to enhance phase separation and translation inhibition (new Fig. 8a-c). We also proved that the gelation of RNP granules is independent of Orb recruitment (better explained below). We are confident that the new information addresses the Reviewer's concern.

2. The mechanism of granule liquid state maintenance by Hecw requires more experiments to understand. It is unclear whether the gel-like transition was caused by the loss of the ubiquitination by Hecw or other potential ubiquitination-independent functions of Hecw. To resolve this question, authors should examine the liquid/gel state of granules in Hecw CI mutant in addition to their existing data from KO flies (Fig. 4 and Fig. 5).

R: We thank the Reviewer for raising this important issue. Following their suggestion, we have now analyzed the liquid/gel state of the granules in Hecw CI mutant flies both in ovaries and in neurons. Me31B::GFP-positive CI RNPs have an irregular shape, are larger in comparison to control and are insensitive to 1,6-hexanediol treatment. Thus, as for Hecw KO, RNPs' morphology and physical properties are altered also in Hecw CI flies, demonstrating that the gel-like transition was caused by the loss of the ubiquitination activity of Hecw. These data are now added in the new Fig. 4 and Fig. 7 and the conclusions obtained are highlighted in the text.

3. If the liquid state is indeed maintained by the ubiquitination, is it due to the ubiquitination of structural components like ME31B or indirectly caused by an accumulation of mis-expressed proteins like Orb? The authors could address this by asking whether ME31B is ubiquitinated by Hecw in vivo and whether reducing Orb by RNAi or using a heterozygote can suppress the gel like state.

R: We thank the Reviewer for bringing this hypothesis to our attention. During the course of our studies, we didn't obtain any evidence of Me31B as possible interactor or substrate of Hecw activity. Nonetheless, considering a possible common regulatory mechanism

exerted by ubiquitin on Me31B, we analyzed the expression level and localization of Bicaudal (BicD), a well-known Me31B direct target {Nakamura, 2001 #26}.

In the panels here reported for the Reviewer's critique, no difference between CTRL and KO, or KI flies are visible with respect to the BicD protein levels measured by immunofluorescence and by immunoblot.

Similar negative observations were made regarding RNPs gel-like state by reducing Orb levels using a heterozygote line (see below).

4. One notable finding of this study is identifying dFMRP as a substrate of Hecw. Consistently, Hecw mutant flies phenocopy dFMRP loss of function mutants in oogenesis. This suggests a novel and intriguing mechanism by which poly-ubiquitination potentiates dFMRP function as a translational repressor. However, proving direct evidence for this conclusions, will require the identification of ubiquitination sites on dFMRP and making the corresponding mutants for functional in vivo study. Alternatively, authors should at least corroborate their Hecw-dFMRP-Orb axis model by testing whether orb heterozygosity (using alleles like orb343 or orbdec) or over-expressing fmr1 can suppress Hecw-KO/CI oogenesis phenotypes.

R: We thank the Reviewer for her/his enthusiasm toward our findings and also for the compelling suggestions, which helped us to improve our manuscript.

To corroborate our findings, we embarked in the genetic validation of the Hecw-dFMRP-Orb axis by bringing both KO e CI Hecw homozygous onto an Orb heterozygous background. The genotypes were then characterized at the immunoblot level and for

oogenesis phenotypes, along with controls, using 3-day-old flies, as shown below. As expected, *Orb* heterozygous lines displayed a reduced *Orb* expression.

Strikingly, in the *Orb* heterozygous background, we could largely rescue the defective egg chambers scored in *Hecw*^{KO} (19%) and *Hecw*^{Cl} (21%) homozygous flies. In fact, *Hecw*^{KO};*Orb*^{343/+} presented only 3% of defective egg chambers (2% of the egg chambers presented encapsulation defects and 0,8% had less than 15 nurse cells) while *Hecw*^{Cl};*Orb*^{343/+} yielded 8% of defective egg chambers (4,2% of the egg chambers presented encapsulation

defects and 3,7% had less than 15 nurse cells). The percentage of aberrant egg chambers in the various mutant flies are reported in the new Supplementary Table 1.

Importantly, these flies were instrumental in definitively disproving the alternative hypothesis suggested by the Reviewer, namely that the aberrant gelation of the RNPs in *Hecw* mutants may be indirectly caused by an accumulation of mis-expressed *Orb*.

When reducing the *Orb* to a level that is no longer visible in the aberrant RNPs granules of the nurse cells, these latter remained morphologically altered and solid-like even after the 1,6-hexanediol treatment. These data are now shown in the new Supplementary Fig. 8.

5. To add more direct support to their conclusion that *Hecw* regulates *Orb*/*Profilin* level specifically through *dFMRP*, the author need to over-express *Hecw* in *fmr1* homozygous mutant background (using viable alleles like *fmr13* or *fmr1Δ113*). *Orb* level should not be decreased by *Hecw* over-expression when *dFMRP* is absent.

R: We agree with the Reviewer. We are very grateful for bringing this issue to our attention. To verify that the effect we scored on *Orb* and *Profilin* levels is mediated by *FMRP*, we followed two alternative strategies. The option suggested by the Reviewer failed, unfortunately, since the *fmr1Δ113* allele lies in the same chromosome of the *UAS-Hecw* transgene and recombination of these two elements was impossible due to their close proximity. On the contrary, we were successful in using a *Fmr1* RNAi fly line. Strikingly, immunoblot analysis highlights that both *Orb* and *Profilin* levels are restored (in ovaries and neurons, respectively) when *Hecw* overexpression is induced in the absence of *Fmrp* (see new Fig. 7e). This additional insight fully substantiates our original conclusions that *Hecw* controls *Orb*/*Profilin* levels through *Fmrp* repressive activity.

Minor points:

1. The genetic nature of the rescuing allele DC504 should be explained in the manuscript or in the Figure 2 legend.

R: We thank the Reviewer for bringing this missing detail to our attention. The rescuing allele is described in the methods section and we now added a note describing the nature of the rescuing allele DC504 in the legend of Fig. 2.

2. Authors noted that both, the defective egg chamber phenotype and the *orb* mis-expression phenotype have limited penetrance (20%-30% egg chambers). How do these two phenotype correlate? Do defective egg chambers tend to have *orb* mis-expression or puncta in nurse cells?

R: This is an interesting point. We could observe the presence of Orb in a mis-specified second oocyte in egg chambers with less than 15 nurse cells, or Orb co-localization with Me31B in egg chambers with supernumerary nurse cells, as in the two representative images below. In a few cases though, mis-specification of a second oocyte could occur also in the absence of a evident nurse cell phenotypes or Orb could be found colocalizing with Me31B in egg chambers with correct oocyte specification and/or nurse cell numbers, so in general phenotypes are not all fully correlating. It is important to note that we have focused our analysis mostly on previtellogenic, stage 5-9 egg chambers, while oocyte specification and germ cell division and encapsulation occur earlier on in the germarium.

3. The *in vitro* ubiquitination experiment presented in Figure 6d needs to include a control lane without *Fmrp* as the poly-ubiquitin chain signal might come from *Hecw* self-ubiquitination and non-specific binding to NiNTA beads.

R: We agree with the Reviewer. To exclude that the signal might come from *Hecw* self-ubiquitination, we used Amylose resin to purify His-MBP-FMRP and we included a control reaction with resin alone to show the minimal non-specific signal coming from self-ubiquitinated *Hecw*

bound to the resin. The left panel represents the pellet, and the right panel shows the supernatants with free Ub chains. The second lane of both panels represents the requested control lane without *Fmrp*; ubiquitination reaction is at completion (no more

monomeric Ub left), yet almost no signal visible in the beads-alone lane. We substituted the former Fig. 6d with the panel showing this data.

4. Fig. 6h and 6i, the authors should note the penetrance of Grk/Osk mis-localization phenotypes.

R: As suggested by the Reviewer, we have now included a detailed quantitation of the phenotype penetrance in Fig. 6h.

Reviewer #3 (Remarks to the Author):

This paper deals with the functional characterization of the Drosophila Hecw ubiquitin ligase during oogenesis and in neurons. The authors generated knockout and catalytically inactive mutant alleles of Hecw and analyzed the phenotypes of the mutants, in particular during oogenesis. The mutants are homozygous viable but show reduced lifespan and impaired climbing behavior, indicative of motoneuron function impairment. A detailed analysis of oogenesis in the mutants revealed that egg chambers displayed a variety of morphological defects, ranging from abnormal number of nurse cells, the presence of two oocytes in one egg chamber, ring canal defects and in general reduced numbers of laid eggs. Intriguingly, the mutants show abnormally sized and shaped RNP granules in germ line cells that contained Orb protein, which is normally confined to the oocyte and is degraded in nurse cells. By use of live imaging analysis and fluorescence bleaching assays the authors could show that the normally liquid-like state of the RNP granules was changed to a more solid, gel-like state in the mutants. Using co-IP and GST binding assays, the authors identified fragile-X mental retardation protein (FMRP), a translational repressor crucial for repression of Orb translation in nurse cells, as a binding partner and ubiquitination target of Hecw. FMRP also has a function in neurons where it regulates the levels of the actin-associated protein profilin. Indeed, profilin levels in Hecw mutant brains were also abnormal, pointing to a general function of Hecw in regulating protein translation via FMRP.

The data in this paper are of excellent quality and the manuscript is well organized and well written throughout. The story told in the manuscript is of general interest and novelty, as RNP granules and the human homolog of FMRP are involved in the pathogenesis of Amyotrophic lateral sklerosis (ALS) a devastating deadly disease. This paper now shows for the first time in the model organism Drosophila that a ubiquitin ligase can regulate the biophysical properties and functions of RNP granules and of the FMRP protein, which may open new avenues in the research on ALS. I recommend publication of the manuscript in Nature Communications, provided that a few minor points will be addressed.

R: We thank the Reviewer for noting the novelty and importance of our work.

Suggestions to the authors:

1. *Supplementary Table 2: This table just lists the names of several proteins that were identified as binding partners of GST-Hecw in a pull-down assay analyzed by mass spectrometry. Here I suggest to publish the full Exel file with proteins identified by mass spec as supplementary file, including significance values, numbers of identified peptides etc., so that the reader can decide on the significance of the potential interaction.*

R: We agree with the Reviewer. As we used to do with our mass spectrometry data, we submitted the raw data to ProteomeXchange via the PRIDE database in order to allow readers to access and independently analyze them. We added a sentence in the manuscript and included the information on how to access the file.

Project Name: Pulldown CG42797 vs S2 cell lysate Project accession: PXD025809

Project DOI: 10.6019/PXD025809

Reviewer account details:

Username: reviewer_pxd025809@ebi.ac.uk

Password: rLv2akd

2. Figure 6: In panels b, h and i immunofluorescence images of oogenesis phenotypes are shown that were observed in FMRP mutants (b) and in Hecw mutants regarding the mislocalization of Gurken (h) and Oskar (i). The frequencies of these phenotypes in comparison to wild type needs to be quantified.

R: We agree with the Reviewer. We have now included a detailed quantitation of the phenotype penetrance in Fig. 6h.

3. Discussion, page 10: The authors discuss that ubiquitin receptors, proteins reading the ubiquitin modifications on proteins such as FMRP, are likely to be involved in the regulation of RNP properties. They mention the ubiquitin receptor Lingerer, which was found in the GST pulldown assay as a potential binding partner of Hecw. They should also mention and discuss the ubiquitin receptor Rings lost (Rngo), which shows strikingly similar oogenesis phenotypes as Hecw mutants in germ line clones (Morawe et al., 2011, J Cell Biol 193, 71)

R: We thank the Reviewer for bringing this publication to our attention, but we are not sure to have understood her/his point. From the published data Rngo seems to act as ubiquitin receptor but in a different context as Rngo appears to be required for ring canal growth and plasma membrane integrity. Maybe the Reviewer is aware of additional unpublished data by the same authors? Anyway, we added a brief sentence in the discussion to cite the paper, as requested.

REVIEWERS' COMMENTS

Reviewer #1 (Remarks to the Author):

My past concerns have been reasonably addressed. The revised paper describes an interesting and important study that should be of general interest. I recommend its acceptance after minor revision.

Minor comments:

1. The authors should give the the full name of ROI.

Reviewer #2 (Remarks to the Author):

We have read the revised manuscript and the authors' rebuttal. The authors have provided new information that addressed our previous concerns. No further changes necessary.

Reviewer #4 (Remarks to the Author):

The data in Figure 2 look interesting and seem to be well-controlled.

One suggestion to improve the impact of the work is to test whether anti-apoptotic genes (e.g., DIAP) can rescue the mutant phenotypes.

This is a suggestion. Not a requirement for publication.

Additional comment after conversation with Reviewer #4:

Please tone down "due to neural loss" to "associated with neural loss" as there is no specific rescue to neurons included in the manuscript.

Reviewer #1 (Remarks to the Author):

My past concerns have been reasonably addressed. The revised paper describes an interesting and important study that should be of general interest. I recommend its acceptance after minor revision.

Minor comments:

The authors should give the full name of ROI.

Reviewer #2 (Remarks to the Author):

We have read the revised manuscript and the authors' rebuttal. The authors have provided new information that addressed our previous concerns. No further changes necessary.

Reviewer #4 (Remarks to the Author):

The data in Figure 2 look interesting and seem to be well-controlled.

One suggestion to improve the impact of the work is to test whether anti-apoptotic genes (e.g., DIAP) can rescue the mutant phenotypes. This is a suggestion. Not a requirement for publication.

Please tone down "due to neural loss" to "associated with neural loss" as there is no specific rescue to neurons included in the manuscript.

R: We are very grateful to all Reviewers for the critical reading of our manuscript throughout the entire process of revision and for the numerous useful suggestions that we have implemented to improve our study. Following their last comments, we have now modified the text adding the full name of ROI and changed the sentence mentioned by Reviewer #4.